# Describing rainfall in northern Australia using multiple climate indices

Cassandra Denise Wilks Rogers[1], Jason Beringer[1,2]

5  [1]School of Earth, Atmosphere and Environment, Monash University, Clayton, 3800, Australia.

[2]School of Earth and Environment (SEE), The University of Western Australia, Crawley, WA, 6009, Australia.

10  *Corresponding author: Jason Beringer, tel. +61 409355496, e-mail: Jason.Beringer@uwa.edu.au*

**Abstract.** Savanna landscapes are globally extensive and highly sensitive to climate change, yet the physical processes and climate phenomena which affect them remain poorly understood and therefore poorly represented in climate models. Both human populations and natural ecosystems are highly susceptible to precipitation variation in these regions due to the effects on water and food availability and atmosphere-biosphere energy fluxes. Here we quantify the relationship between climate phenomena and historical rainfall variability in Australian savannas, and in particular, how these relationships changed across a strong rainfall gradient, namely the North Australian Tropical Transect (NATT). Climate phenomena were described by 16 relevant climate indices and correlated against precipitation from 1900 to 2010 to determine the relative importance of each climate index on seasonal, annual and decadal time scales. Precipitation trends, climate index trends, and wet season characteristics have also been investigated using linear statistical methods. In general, climate index-rainfall correlations were stronger in the north of the NATT where annual rainfall variability was lower and a high proportion of rainfall fell during the wet season. This is consistent with a decreased influence of the Indian-Australian monsoon from the north to the south. Seasonal variation was most strongly correlated with the Australian Monsoon Index, whereas yearly variability was related to a greater number of climate indices, predominately the Tasman Sea and Indonesian sea surface temperature indices (both of which experienced a linear increase over the duration of the study) and the El Niño-Southern Oscillation indices. These findings highlight the importance of understanding the climatic processes driving variability, and subsequently the importance of understanding the relationships between rainfall and climatic phenomena in the Northern Territory in order to project future rainfall patterns in the region.

## 1 Introduction

Human populations and ecosystems are highly susceptible to changes in both intensity and timing of precipitation events through their impact on global energy, water and carbon cycling (Beringer et al., 2011b). Knowledge of atmospheric processes and circulation patterns are important for assessing rainfall patterns under both natural variability and anthropogenic climate change. However, precipitation remains relatively poorly modelled at a global scale due to the inability of coupled Atmosphere-Ocean General Circulation Models to adequately model tropical rainfall and cloud formation (Randall et al., 2007). Possible changes to the El Niño-Southern Oscillation (ENSO) system due to climate change and inconsistencies in model simulations of precipitation related to the Indian-Australian monsoon limit our ability to project future precipitation changes in northern Australia with much confidence (Christensen et al., 2007). Global precipitation patterns and rainfall trends over time are driven by many different climatic phenomena and interactions between the atmosphere, ocean and land surface over different timescales (Wallace and Hobbs, 2006). Major atmospheric and oceanic circulation patterns and systems are some of the major drivers of Australian rainfall, with ENSO (e.g. Nicholls, 1988), the Indian Ocean Dipole (IOD, e.g. Risbey et al., 2009), the Indian-Australian monsoon (also known as the Asian-Australian monsoon, e.g. Kumar et al., 1999) and synoptic-scale weather events in the extra-tropics (Sturman and Tapper, 2006) among those known to have the greatest influence on Australian rainfall.

Climatic phenomena, such as the Indian-Australian monsoon, affect rainfall and water balance on a seasonal scale (Sturman and Tapper, 2006). On longer time scales, natural climatic oscillations can cause inter-annual (e.g. ENSO) to decadal scale (e.g. the Pacific Decadal Oscillation) variability (Kamruzzaman et al., 2013). Ultimately rainfall is a complex part of the climate system that is dependent on many different factors, including various climatic phenomena, over many different time

scales (Risbey et al., 2009).

One common method used to research the influence of climatic phenomena on various climatic variables, such as rainfall, is through the use of climate indices (e.g. Ashok et al., 2007; Kamruzzaman et al., 2013; Schepen et al., 2012). A climate index is a numerical value which provides a measure of the change away from the mean in an oscillatory climate system (National

Center for Atmospheric Research, 2012). Climate indices generally consist of a single value, such as a measure of sea surface temperature (SST), mean sea level pressure (MSLP) or wind speed, which can be related via statistical analysis to highly complex climatic processes. For example, Australian rainfall is well correlated with the Southern Oscillation Index (SOI) which is in turn associated with ENSO (e.g. Murphy and Ribbe, 2004; Risbey et al., 2009; Schepen et al., 2012).

There remain many uncertainties surrounding the impacts that anthropogenic climate change will have on precipitation, which Rowell (2011) argues is because rainfall model uncertainty in Australia is due to natural variation rather than errors in climate models or observational data. For example, in Queensland, Murphy and Ribbe (2004) found the relationship between ENSO and rainfall did not remain constant over time. They ascribed this to the change in the status of the Inter-decadal Pacific Oscillation. It is therefore important to better understand natural variation in circulatory climate systems to improve future

climate projections. Over longer time periods, relationships between climatic phenomena may strengthen or weaken due to climate change. One potential major impact on Australian rainfall is an increase in northern Australian wet season rainfall due to an intensification of the monsoon. Kumar et al. (1999) observed an intensification of the monsoon due to a weakening in its relationship with ENSO. Climate change is also expected to alter other atmospheric circulation systems, such as the weakening of the Walker circulation (Power and Smith, 2007; Vecchi and Soden, 2007) and the broadening of the Hadley Cell

(Brönnimann et al., 2009), making the potential impact of climate change on precipitation in tropical Australia very uncertain. Understanding the relationship between climate drivers and rainfall under both natural variability and anthropogenic change is required to project the future impact of climate change on food, fibre and water production in savanna regions.

In Australia, rainfall variability is crucial to the structure and productivity of the landscape, particularly across the vast extent

of the savanna biome which accounts for 25% of the continent (Beringer et al., 2007, Beringer et al., 2011b; Eamus et al., 2013; Haverd et al., 2016; Hutley et al., 2011; Kanniah et al., 2011). Not only do savanna landscapes cover a large portion of Australia, they account for 15% of the land surface of the planet (Beringer et al., 2011b). This area has the potential to increase in size due to climate change (Franchito et al., 2011), making the improved knowledge of these landscapes important in

understanding future rainfall trends in Australia and around the world. Australian savannas remain fairly undisturbed (Beringer et al., 2014) making them good for temporal change studies. Vegetation productivity, and hence the carbon balance, are vulnerable to changes in rainfall variability (Kanniah et al., 2011) because savanna structure, composition and function shift in response to short (monsoonal) and long term (ENSO, Inter-decadal Pacific Oscillation, Pacific Decadal Oscillation Index

(PDO), etc.) rainfall climatology (Beringer et al., 2011a). Moreover, disturbances, such as fire, cyclones and grazing, are also key drivers of savanna structure and productivity which are in turn driven by rainfall patterns (Beringer et al., 2007; Bond et al., 2003; Hutley and Beringer, 2011; Hutley et al., 2013). Savanna grass productivity is very sensitive to rainfall and the biomass produced creates fodder for cattle and fuel for frequent burning, resulting in greenhouse gas emissions (Beringer et al., 1995, 2014; Moore et al., 2015, 2016). Therefore, evaluating the relationship between inter-annual variation in rainfall and

climate phenomena is crucial for predicting the responses of the water, energy and carbon cycles of savanna vegetation (Beringer et al., 2011a; Kanniah et al., 2013). Despite this there has been a paucity of research undertaken on the climatic influences of rainfall in northern Australia and savannas, however, there have been a number of continental scale analyses (e.g. Nicholls, 1989; Risbey et al., 2009; Schepen et al., 2012).

From north to south along the Northern Territory (NT), there is a substantial rainfall gradient (Table 1 and Fig. 1) known as the North Australian Tropical Transect (NATT). The north is highly seasonal with a characteristic tropical monsoonal climate (Bureau of Meteorology, 2011b; Hutley et al., 2011) and a rainy season between September and May (Nicholls et al., 1982; Suppiah and Hennessy, 1996), whereas the south is semi-arid to arid (Beringer et al., 2016) with very little seasonal variation in rainfall (Hennessy et al., 1999). A sharp change in rainfall rate occurs around Dry Creek, with the stations north of Dry

Creek experiencing higher rainfall, and the stations further south experiencing less rainfall.

Due to humanity's heavy demand for reliable and plentiful sources of water, combined with the threat of unknown future rainfall changes due to anthropogenic climate change, there is a need to improve our knowledge of precipitation drivers in order to improve future rainfall projections. The high importance of the Australian savanna landscapes to the global carbon

and water balances illustrates the need to better understand the climatic processes in these regions. This paper determines how well different climate indices describe rainfall along the NATT. This has been achieved using Pearson product-moment correlations. This research addresses correlation, not causation, over an extensive historical record (1900 to 2010). Trends in climate indices and rainfall over the NATT for this time period have also been examined. An unprecedented number of climate indices, representing climatic phenomena that are known to influence Australian rainfall variability, have been implemented

in this research.

## 2 Methodology

### 2.1 Site description

This paper examines the strong rainfall gradient along an extended version of the NATT, where 16 sites were chosen to examine the spatial relationships of rainfall with 16 different climate indices (Table 1 and Fig. 1). Due to the recognition of

the importance of the NATT as a 'living laboratory' (Hutley et al., 2011) and the role of rainfall is ecosystem structure and function (Beringer et al., 2011b), a number of micrometeorological flux measurements have been established along the NATT as part of the regional flux network (OzFlux). Beringer et al. (2016) provide a description of the OzFlux network with initial cross site analysis. Howard Springs has been a long term monitoring site with observations initiated in 1996 (Eamus et al., 2001) and other sites include Fogg Dam (Beringer et al., 2013), Daly River (Hutley et al., 2011), Dry Creek (Beringer et al.,

2011b) and Alice Springs (Cleverly et al., 2013). Rainfall decreases along the NATT from approximately 1,600 mm per year in the north, at a rate of ~200 mm per degree, to approximately 200 mm per year in the south, at a rate of ~100 mm per degree (Bureau of Meteorology, 2011a; Cook and Heerdegen, 2001) (Table 1 and Fig. 1). Associated with this decrease in precipitation is a change in vegetation structure and composition which varies from moist woodland savanna in the north to dry grasslands in the south (Beringer et al., 2011b; Hutley et al., 2011). Seasonality of rainfall also decreases from the north

to the south, mostly due to the decrease in the influence of the monsoon moving further inland (Cook and Heerdegen, 2001).

### 2.2 Climate indices

We used an extensive 110 years of spatial data, from 1900 to 2010, to help identify persistent long term correlations between precipitation and climate indices and to capture multiple events of each climatic phenomenon occurring at different frequencies. We also advanced on previous studies by examining a vast number of climate indices (16) as described below

and summarised in Table 2. The use of multiple climate indices enabled us to gain an insight into which climatic phenomena may have the strongest relationships with spatial rainfall patterns in the NT, and ultimately which climatic phenomena may have an effect on ecosystem structure, function and distribution. A map showing the regions over which each climate index is calculated is shown in Fig. 2.

Indian Ocean and Indonesian phenomena
Climatic phenomena over the Indian Ocean and Indonesia are related to Australian precipitation both directly (e.g. Kamruzzaman et al., 2013) and indirectly by influencing other climatic phenomena (e.g. the Inter-decadal Pacific Oscillation and ENSO (Power et al., 1999)). SST anomaly (SSTA) data over these regions were used to calculate four climate indices as follows (Table 2 and Fig. 2). The IOD, represented by the _Dipole Mode Index_ (DMI), is defined as the difference between the

_Indian Ocean West Pole Index_ (WPI, the average of the SSTAs over 50°E to 70°E and 10°N to 10°S) and the _Indian Ocean East Pole Index_ (EPI, the average of the SSTAs over 90°E to 110°E and 0°N to 10°S (Saji et al., 1999)). Changes in the DMI

coincide with changes in equatorial zonal wind variation (Saji et al., 1999). Anomalies in the DMI begin around June and increase until October when they reach a maximum, after which they quickly return to normal (Saji et al., 1999).

The *Indonesia Index* (II, the average of the SSTAs over 120°E to 130°E and 0°N to 10°S) characterises SSTAs over the Indonesian region and has been related to eastern Australian winter rainfall (Verdon and Franks, 2005) and NT rainfall
(Schepen et al., 2012).

El-Niño Southern Oscillation

The relationship between ENSO and Australian rainfall is well known (e.g. Risbey et al., 2009; Ropelewski and Halpert, 1996). There are multiple ENSO indices of which we used six (Table 2 and Fig. 2). The *Southern Oscillation Index* (SOI) is commonly
used in Australian rainfall studies (e.g. Risbey et al., 2009; Ropelewski and Halpert, 1996; Schepen et al., 2012; Suppiah and Hennessy, 1996). The SOI for a given month is calculated as a function of MSLP difference between Tahiti and Darwin (Bureau of Meteorology, 2012).

The four *Niño indices* represent ENSO using SSTA measurements that are averaged over different regions of the Pacific Ocean (Fig. 2). These indices and their corresponding regions are; the Extreme Eastern Tropical Pacific SST (Niño 1+2), 90°W to
80°W and 0°N to 10°S; the Eastern Tropical Pacific SST (Niño 3), 150°W to 90°W and 5°N to 5°S; the East Central Tropical Pacific SST (Niño 3.4), 170°W to 120°W and 5°N to 5°S; and the Central Tropical Pacific SST (Niño 4), 160°E to 150°W and 5°N to 5°S (ESRL, 2012; Kamruzzaman et al., 2013; Risbey et al., 2009). Kamruzzaman et al. (2013) noted a seasonal pattern in Niño 1+2 and Niño 3 from 1957 to 2007.

The *El Niño Modoki Index* (EMI) quantifies El Niño-Southern Oscillation Modoki events, which are similar to traditional
ENSO events (Ashok et al., 2007), but with the maximum warming further east than normal (Risbey et al., 2009). The EMI is defined by the following equation; EMI = C – 0.5 * (E + W), where C represents SSTAs over 165°E to 140°W and 10°N to 10°S, E represents SSTAs over 110°W to 70°W and 5°N to 15°S and W represents SSTAs over 125°E to 145°E and 20°N to 10°S (Ashok et al., 2007; Schepen et al., 2012).

Extra-tropical phenomena

The *Tasman Sea Index* (TSI), defined as an area off the east coast of Australia bounded by 150°E to 160°E and 30°S to 40°S (Murphy and Timbal, 2008), was included in this research to investigate if there is a potential link between extra-tropical SSTs and NT rainfall (Table 2 and Fig. 2). We used the average of the SSTAs over this area to calculate the TSI. The use of the TSI to quantify extra-tropical SSTs has only been used in recent years (e.g. Murphy and Timbal, 2008; Schepen et al., 2012) so
there are not many studies that include this index.

Atlantic and tropical phenomena

Links have been found between Atlantic Ocean SSTs and rainfall in south-eastern (Kamruzzaman et al., 2013) and north-western Australia (Lin and Li, 2012). The _North Atlantic Index_ (NATL, average SSTA over 60°W to 30°W and 20°N to 5°N) and the _South Atlantic Index_ (SATL, average SSTA over 30°W to 10°E and 0°N to 20°S (National Oceanic and Atmospheric Administration, 2012)) have been included in this research to determine if there is any potential teleconnection between NT rainfall and Atlantic Ocean SSTs (Table 2 and Fig. 2). Kamruzzaman et al. (2013) noted the presence of a seasonal pattern in the NATL and the SATL as well as an increasing trend in both from 1957 to 2007.

SSTs from the tropical regions in the Pacific, Atlantic and Indian Oceans all have links with Australian rainfall (Lin and Li, 2012; Risbey et al., 2009; Schepen et al., 2012). The _Global Tropics Index_ (TROP, the average of the SSTAs between 10°N and 10°S around the equator (National Oceanic and Atmospheric Administration, 2012)) incorporates all three of the above mentioned oceans into one climate index (Table 2 and Fig. 2). Correlations have been found between south-eastern Australian rainfall and TROP (Kamruzzaman et al., 2013). Kamruzzaman et al. (2013) noted the presence of a seasonal pattern in TROP as well as a quadratic trend from 1957 to 2007.

Indian-Australian Monsoon

The Indian-Australian monsoon is highly influential on tropical Australian rainfall (Sturman and Tapper, 2006) and is characterised by the reversal of easterly trade winds in the Australian tropics (Hendon et al., 2012). Kajikawa et al. (2009) reason that Indian-Australian monsoon variability can be depicted using a measurement of average zonal wind (U-wind) velocity at 850 mb over 110°E to 130°E and 5°S to 15°S, known as the _Australian Monsoon Index_ (AUSMI, Table 2 and Fig. 2), however the definition used for this research differed slightly (see Section 2.3.3). AUSMI is negative for the majority of the year, when the easterly trade winds are dominant, until the start of the monsoon when these winds weaken and then become positive. Kajikawa et al. (2009) found AUSMI to be a good predictor of variability in the monsoon on seasonal, intra-seasonal, inter-annual and inter-decadal time scales.

The Pacific Decadal Oscillation

The _Pacific Decadal Oscillation Index_ (PDO) is a measure of the Pacific Decadal Oscillation, a shift in Pacific Ocean temperatures that has a period of 20 to 30 years (Joint Institute for the Study of the Atmosphere and Ocean, 2012). The Pacific Decadal Oscillation is known to affect rainfall and global temperatures (Franks, 2002; Kamruzzaman et al., 2013). The PDO is defined as the leading principal component of SSTs in the north Pacific (Joint Institute for the Study of the Atmosphere and Ocean 2012) (Table 2 and Fig. 2). The causes of the Pacific Decadal Oscillation remain unknown, therefore it is not possible to predict changes in the PDO (Joint Institute for the Study of the Atmosphere and Ocean, 2012). This limits the usefulness of the PDO for forecasting climate variability. Note, the Inter-decadal Pacific Oscillation was not included in this research as it is highly correlated with the Pacific Decadal Oscillation (Franks, 2002).

## 2.3 Data description and sources

All climate index data were calculated using monthly data whereas rainfall data were aggregated from daily to monthly data. Climate index and rainfall data were also aggregated from monthly to seasonal and annual timescales as appropriate. Throughout this research various time periods will be referred to, including summer, autumn, winter and spring, which all use austral definitions (i.e. December 1 to February 28 or 29 depending on the year, March 1 to May 31, June 1 to August 31 and September 1 to November 30 respectively). Unless otherwise stated, rainfall data will be reported as daily averages to remove any inconsistencies created by differing season or month lengths. We defined a year to start on July 1 and end on June 30 (the hydrological year) so that each wet season (as defined in Section 2.7) is not split over two calendar years. It is important to note that none of the data used for this study have been detrended and the seasonal cycle has not been removed.

## 2.3.1 Precipitation data

Australian gridded daily rainfall data, interpolated from station data (1900 to 2010), were obtained from the Australian Water Availability Project (AWAP) (Raupach et al., 2009, 2012), which was developed by the Commonwealth Scientific and Industrial Research Organisation (CSIRO). AWAP resolution is 0.05° latitude by 0.05° longitude, approximately 5 km by 5 km (CSIRO 2012). While AWAP data were available nationally from 1900 to 2010, at specific sites there were a very small number of missing values (~ 0.1% at the NATT sites). At the NATT sites, in the case of a single missing point, data were linearly interpolated. For consecutive missing values, daily averages for each month (excluding months with missing data) were calculated for each location and these averages replaced the missing values for the corresponding month.

## 2.3.2 Sea surface temperature indices and ENSO data

PDO data were obtained from the Joint Institute for the Study of the Atmosphere and Ocean (JISAO), Washington University (http://jisao.washington.edu/pdo/PDO.latest). SSTA data for the WPI, EPI, DMI, TSI, Niño1+2, Niño3, Niño3.4, Niño4, NATL, SATL, TROP, EMI and II were extracted from the Extended Reconstruction Sea Surface Temperature version 3b (ERSST.v3b) via the National Oceanic and Atmospheric Administration (NOAA) – National Operational Model Archive and Distribution System (NOMADS) Live Access Server (LAS) (http://nomads.ncdc.noaa.gov/las/getUI.do). ERSST.v3b is based on the International Comprehensive Ocean-Atmosphere Data Set (ICOADS) release 2.4 (National Climatic Data Center, 2012). For more information about past ERSST versions see Smith et al. (2008b). ERSST.v3b produces monthly SST and SSTA data over 2° grid boxes (Xue et al., 2010) using buoy and ship observations (National Climatic Data Center, 2012; Smith et al., 2008b; Yates et al., 2008). Data obtained from ERSST.v3b may be prone to uncertainty created by incomplete sampling or data errors (Smith et al., 2008b). Anomaly data are then created using a base period climatology from 1971 to 2000 following Xue et al. (2010). Monthly SSTA data were available from the 1854 to 2012 reconstruction, however data before 1940 are considered less reliable (NCDC, 2012). The use of SSTAs for some of these indices is consistent with other climate index research, e.g. WPI, EPI, DMI, Niño3, Niño3.4, Niño4, EMI, II and TSI by Schepen et al. (2012) and the Niño Indices, SOI,

PDO, DMI, NATL, SATL and TROP by Kamruzzaman et al. (2013). SOI data were obtained from the Australian Bureau of Meteorology (http://www.bom.gov.au/climate/current/soihtm1.shtml).

### 2.3.3 Monsoon index data

The definition used for AUSMI in this research is slightly different to that used by Kajikawa et al. (2009) and was instead defined as the 850 mb zonal wind velocity averaged over the area enclosed within 110°E to 130°E and 6°S to 14°S. Eastward zonal wind velocity (m s$^{-1}$) data were extracted from the NOAA- Cooperative Institute for Research in Environmental Sciences (CIRES) 20th Century Reanalysis version 2 (20CR): Monthly Mean Pressure Level Data at 850 mb. The 20CR provides reanalysed weather data over time and space from the late 1800s to the present (http://www.esrl.noaa.gov/psd/data/gridded/data.20thC_ReanV2.pressure.mm.html). The 20th Century Reanalysis Data were made available by the NOAA/Office of Ocean and Atmospheric Research (OAR)/ Earth Systems Research Laboratory: Physical Sciences Division (ESRL, 2012), Boulder, Colorado, USA. The data were extracted for the latitudes 6, 8, 10, 12 and 14°S; and for the longitudes 110, 112, 114, 116, 118, 120, 122, 124, 126, 128 and 130°E. Since the data were only available for every 2 degrees of latitude it was not possible to be exactly consistent with Kajikawa et al. (2009). Once the data were extracted they were averaged over the specified region to produce a one dimensional time series.

### 2.4 Trends

Before we assessed correlations between precipitation and climate indices we examined for linear trends in both the rainfall data and the climate indices. Linear regression and smoothing were used to identify persistent linear trends between 1900 and 2010.

### 2.5 Correlation maps

Maps of the correlations between rainfall and climate indices were created using Pearson product-moment correlation coefficients of determination (r$^2$ values) to quantify the correlations between rainfall and climate indices for every grid point (5 km resolution) across Australia using the gridded meteorological AWAP data. Significance was assessed through p values at the 95% significance level and only significant data were plotted throughout this research. Maps of r$^2$ and p values were used to determine which climate index was correlated best with rainfall at each grid point over Australia. These maps were spatially aggregated from 0.05° x 0.05° to 1.25° x 1.25° to aid visualisation (e.g. Fig. 5 and 6). The spatially aggregated maps illustrated which climate index was most strongly correlated with rainfall, not how strongly it was correlated. Correlation maps were created for concurrent monthly and annual rainfall correlations for each of the 16 climate indices. Even though the focus of this research was on the NATT and the NT, correlations over the whole of Australia are shown to put the regional correlations in the broader Australian context.

## 2.6 Point correlations

Correlation strengths between rainfall and each climate index were determined for every site along the NATT (Table 1 and Fig. 1) for monthly, seasonal and annual time periods. These correlations are referred to throughout the paper as *point correlations*. The point correlations allowed for an in depth analysis of which climate indices were most highly correlated with rainfall along the rainfall gradient (i.e. NATT) by providing a visual comparison of the relative correlation strengths of multiple indices in two dimensions.

For the purposes of this research, $r^2$ greater than 0.6 was considered very strong, between 0.2 and 0.6 was considered strong, between 0.1 and 0.2 was considered moderate and less than 0.1 was considered weak. These definitions were based on analysis done by Murphy and Ribbe (2004), Suppiah (2004) and Suppiah and Hennessy (1996) and were mainly intended to be used to compare relative correlation strengths between different indices, rather than to determine a definitive measure of strength. To compare point correlation strengths between climate indices over each time period, a numerical rank, referred to in this research as an index rank, was calculated. Each climate index was given a rank from 1 (strongest correlation) to 16 (weakest correlation) for each site along the NATT. The average rank over all sites for each climate index was then used to find which indices showed, on average, the strongest correlations along the entire NATT.

## 2.7 Wet season characteristics

Wet season start (onset) date was defined as the date when rainfall, between September 1 and April 30 the following year, exceeded 15% of the rainfall total between these two dates (Smith et al., 2008a). This definition was created by Smith et al. (2008a) who cited a similar definition by Nicholls (1984). Wet season end (retreat) date was defined as the date when rainfall, between September 1 and April 30, exceeded 85% of the total rainfall (Smith et al., 2008a). The time between the start and end date is referred to as the wet season. Wet season onset and retreat is not to be confused with the monsoon onset and retreat. During the wet season the tropics experience heavy precipitation. Similarly, the monsoon is characterised by heavy rainfall but, unlike the wet season, the monsoon is also characterised by shifts in the wind and pressure in the region. Any single wet season typically consists of multiple monsoon active and break phases (Suppiah and Hennessy, 1996) but these have not been examined in this research. The wet season definition used for this research was chosen as it should cover all monsoon active phases and ensure that both start and end dates exist for each site along the NATT for every year. Values were calculated for mean total wet season rainfall, maximum and minimum total, average wet season start and end date, average duration, number of rain days (number of days during the wet season where rainfall is greater than zero) and rainfall intensity for each NATT site.

## 2.8 Limitations

One limitation of this study was the lack of inclusion of indices which measure the Madden-Julian Oscillation and the Southern Annular Mode. An area of further research for this study could include the inclusion of these phenomena but this may require significantly shortening the study period. Data accuracy in observational, reconstructed and modelled data all have implications on the correlations determined in this research. The length of this study poses some issues due to changes in observational equipment used to measure environmental variables and reduced spatial coverage of recording stations or gauges in the past. For example, ERSST.v3b data quality is compromised by a cold bias evident in historical SST data around the 1940s. This bias is due to changes in observational methods associated with World War II (Smith and Reynolds, 2003). Additionally, AWAP data accuracy is limited by the sparseness of observation points (CSIRO, 2012).

## 3 Results and discussion

### 3.1 Trend analysis

Linear trends in rainfall and each climate index were analysed over the full 110 year time period. Fractional rainfall change was calculated for each site over the NATT, revealing a significant increase in rainfall at all sites (Table 3). Hennessy et al. (1999), Li et al. (2012) and Nicholls (2006) all noted similar rainfall trends from 1910 to 1995, 1948 to 2007 and 1950 to 2005 respectively, however only Li et al. (2012) found the increase was significant. While the increase in rainfall in the north was mostly greater in magnitude than in the south, rainfall change in the south was more variable and less linear (Table 3). The dominant influence of the monsoon on northern rainfall (Sturman and Tapper, 2006) is likely to explain these observations of generally high summer and low winter rainfall in the north, whereas the low rainfall in the south has many different and inconsistent influences (Beringer and Tapper, 2000; Sturman and Tapper, 2006; Suppiah and Hennessy, 1996).

We found a significant increase in the magnitude of most climate indices (WPI, EPI, TSI, Niño1+2, Niño3, Niño3.4, Niño4, NATL, SATL, TROP, and II) and a significant decrease in AUSMI. All other indices showed no linear trend. The significant increases in SSTA data are consistent with conclusions in the Fourth Assessment Report by the Intergovernmental Panel on Climate Change (IPCC) that oceanic temperatures have increased (Bindoff et al., 2007). In this study we did not assess long term changes in the frequency or intensity of climate indices or changes in relationships between different climate indices. However such changes have been documented, for example Vecchi and Soden (2007) noted a weakening in the Walker circulation while Power and Smith (2007) noted a reduction in the magnitude of the SOI and the possible dominance of an El Niño pattern from 1977 to 2006. Kumar et al. (1999) noted a weakening in the relationship between ENSO and the monsoon, leading to an intensification of the monsoon, and Nicholls et al. (1996) observed a reduction in the SOI due to a change in the relationship between Australian rainfall, temperature, and the SOI.

AUSMI showed a fairly weak but significant negative linear trend over the study period. This finding suggests a weakening of the monsoon or a reduction in the duration of time when the monsoon is active. The observed decrease in AUSMI appears to contradict the observed precipitation increase over the NATT (Table 3), however the reduction in AUSMI is not evidence that the monsoon has weakened, rather that its circulation characteristics may have changed. Rainfall was also observed to have increased in northern Western Australia which has been linked to an increase in aerosol emissions originating from Asian population centres (Rotstayn et al., 2007). This may help to explain the trends along the NATT but, as noted by Rotstayn et al. (2007), more research is required before a link between Asian aerosols and Australian rainfall can be determined.

It is important to note that since we have not detrended rainfall or climate index data, relationships between rainfall and any of WPI, EPI, TSI, Niño1+2, Niño3, Niño3.4, Niño4, NATL, SATL, TROP, II and AUSMI (all of which show a linear trend) may be due to coincident trends rather than any physical relationship between the given climatic phenomenon and rainfall. Any relationships between rainfall and the climate indices that did not show any linear trends (DMI, SOI, EMI and PDO) are less likely to be coincidental.

### 3.2 Seasonality

Northern Australia and the north of the NATT have a highly seasonal climate. The wet season, as defined in the methods section, accounted for between 60% and 74% of annual rainfall along the NATT from 1900 and 2010. The fact that most precipitation occurred during this period demonstrates the importance of understanding wet season dynamics. Both the number of rain days and the intensity of rainfall on these days showed a decreasing non-linear trend from the north to the south of the NATT.

The wet season start day was earliest in the south of the NATT and latest in the mid-north (i.e. SP site) (Fig. 3), however the wet season start day has previously been found to occur earliest in the north (Smith et al., 2008a). This inconsistency is likely due to the use of a percentile based definition of onset and retreat dates in this research, rather than using a definition based on the location of the monsoon trough, as also noted by Smith et al. (2008a) in a study examining similar wet season characteristics. Our choice of a percentile based definition for the wet season is therefore not perfect but it allows onset and retreat dates to be defined every year, unlike some more traditional definitions, and is quite simple to calculate (Smith et al., 2008a). Wet season end day showed much less variation than the start day and generally occurred earlier in the south (Fig. 3). The duration of the wet season was shortest in the northern half of the NATT and increased southward. Beyond about 19° south, rainfall became more variable and less seasonal, producing earlier and longer calculated wet seasons (Fig. 3). The annual mean wet season rainfall volume decreased from the north to the south of the NATT (Fig. 4). Rainfall in the north was normally distributed whereas in the south it showed a positive skew. Standard deviations were greatest in the north whereas coefficients

of variability were greatest in the south, showing that precipitation variability was greatest in the south with respect to mean annual rainfall.

From the above discussion it is clear that relative variability in wet season total, start and end date, duration, number of rain days and intensity were all greater in the south of the NATT. These findings are consistent with the earlier results that annual relative rainfall variability was greatest in the south. The lower variability in the north was most likely due to the dominance of the monsoon, creating relatively consistently high rainfall every year, whereas as discussed earlier, rainfall in the south was affected by more factors, the strength of which vary from year to year such as incursion of cold fronts (Beringer and Tapper, 2000).

## 3.3 Rainfall-climate index correlations

Correlations between rainfall and climate indices were determined along the NATT (Fig. 7 and 8), and over Australia using gridded AWAP rainfall data at a 5 km resolution. Maps are displayed at a 125 km resolution for clarity of viewing (Fig. 5 and 6). The highest statistically significant correlations, on both seasonal and yearly time scales, were found between rainfall and SOI, AUSMI, II, TSI, EMI, and to a lesser degree, EPI (Tables 4 and 5; Fig. 5, 6, 7 and 8). The relatively high correlations between these indices and rainfall indicates there may exist teleconnections between NT rainfall and Indonesian, Tasman and Pacific SSTs and the climate phenomena that affect these regions, mainly the monsoon, ENSO and the IOD, but further research would be required to determine any mechanistic links, particularly as these relationships may be coincidental due to the linear trends in rainfall, AUSMI, II, TSI and EPI. While most climate indices chosen for this study showed some significant correlations with rainfall, some did not; these were SATL (except for correlations with annual data), NATL, TROP, and PDO. This finding suggests that NT rainfall may not show a concurrent, linear relationship with Atlantic and tropical SSTs, north Pacific decadal SST variability, and the climate phenomena that affect these regions.

### 3.3.1 Correlations between monthly rainfall and climate index data

The highest correlations between rainfall and any climate index were found at the monthly time scale (Fig. 7), particularly in the north. The most highly correlated index for monthly data was AUSMI, followed by SOI and EMI (Fig. 8a and Table 5). Correlation coefficients between AUSMI and rainfall were very strong in the north of the transect, but weak in the south. The AUSMI-rainfall correlation gradient was negative, very strong and highly linear ($r^2 = 0.98$) (Fig. 8a), reflecting the strong gradient in rainfall seasonality from the north to the south. On the other hand, this relationship could be largely due to linear trends in both AUMSI and rainfall, of which absolute rainfall trends were greatest in the north, but relative rainfall trends were greatest in the south. Correlations between rainfall and both SOI and EMI were consistently weak along the entire NATT (Fig. 8a). AUSMI was the most highly correlated climate index spatially over the greater northern Australian region (Fig. 5). AUSMI was also correlated over a large portion of southern Australia, including almost all of Tasmania (the southern-most region of

Australia, Fig. 5), suggesting this relationship is an artefact since the monsoon is known to not extend this far south (Bureau of Meteorology, 2008). The seasonal wind shift associated with AUSMI is linked to the wet summer and dry winter seasons in tropical Australia, which coincides with the wet winters and dry summers respectively in temperate southern Australia. Thus, the correlation between AUSMI and southern Australian rainfall was most likely due to seasonal variation and the southern passage of the sun during autumn and the northern passage during spring, rather than a teleconnection between the monsoon and southern Australian rainfall. AUSMI is capturing the annual periodicity associated with the seasonal progression of the sun each year. While the monsoon may in fact have the strongest influence on rainfall variability in the north of the NT, from Fig. 5 it is not possible to determine how far south this possible relationship extends. We must therefore use caution when interpreting the results of this analysis. The AUSMI-rainfall correlation gradient was negative, very strong and highly linear ($r^2 = 0.9850$) (Fig. 8a), reflecting the strong gradient in rainfall seasonality from the north to the south.

### 3.3.2 Correlations between yearly rainfall and climate index data

To examine annual to decadal correlations, annual rainfall data were examined. The three climate indices with the strongest correlations with annual rainfall over the NATT were TSI, II, and SOI, all of which had similar correlation strengths (Fig. 6 and 8b). In general, correlations were moderate to strong in the north and weak to moderate in the south showing that rainfall variability became less well described by these climate indices southwards along the NATT (Fig. 8b).

The fact that TSI and rainfall were more strongly correlated in the north than the south of the NATT (Fig. 8b) was surprising since the TSI region of measurement is geographically closer to the southern end of the transect (see Fig. 2). Frontal weather systems are likely to directly affect both the TSI region of measurement and the southern NATT, but not the northern NATT, further suggesting that we would expect TSI to be more strongly related to southern NATT rainfall. Our results are not consistent with Schepen et al. (2012) who found TSI-rainfall correlations, using a one month lag, to be greatest in the south-west or the centre of the NT between 1950 and 2009 between October and January. Our research suggests that the correlations persisted for the entire year as we found significant correlations using annually averaged data (Fig. 8b). Sturman and Tapper (2006) noted a strong relationship between monsoon onset and synoptic weather in the extra-tropics. This relationship may help to explain the high correlations between TSI and NATT precipitation. Although, as mentioned earlier, both the TSI and rainfall have a significant linear trend which may produce higher correlations than are realistic.

One hypothesis to explain the correlations between Tasman Sea SSTs and NT rainfall is the link both have with Rossby waves (planetary scale atmospheric waves that circle the South Pole) and extra-tropical weather events. Rossby waves are known to have a strong relationship with extra-tropical weather events including high and low pressure cells (Kump et al., 2010; Sturman and Tapper, 2006) and display coupling with global SSTs (Hill et al., 2000). Therefore, the TSI-NATT rainfall correlations

could be due to the simultaneous effects of Rossby waves on both extra-tropical weather, which are known to affect NT rainfall (Sturman and Tapper, 2006) and Tasman Sea SSTs. Additional research is necessary to further investigate this hypothesis.

### 3.3.3 Annual relationships by season

To examine the relationships between seasonal rainfall and climate indices, data were averaged over each season for each year, then rainfall-climate index correlation strengths were evaluated for each season. On a seasonal scale, the strong correlations between AUSMI and rainfall observed in the monthly data persisted during summer and autumn (Tables 4 and 5), which is consistent with the well-known influence of the summer monsoon on northern Australian rainfall (Bureau of Meteorology, 2008; Suppiah and Hennessy, 1996). Spring rainfall showed significant correlations with a large number of climate indices across most sites, unlike the other seasons, and also showed the highest number of sites with significant correlations. Significant correlation strengths ranged from strong (maximum $r^2 = 0.31$) in the north of the transect, to moderate to weak correlations ($r^2 < 0.02$) in the south. This high number of significant correlations at the highly seasonal northern sites may be due to potential relationships between monsoon onset and various climate indices. On the other hand, climate index-rainfall correlations in the south could be due to extra-tropical convection in the southern part of the NATT.

Interestingly the dominance of the summer and spring correlations in the north of the transect is replaced by stronger correlations with winter rainfall south of 23° latitude (Fig. 7). During most seasons, the monsoon and the location of the sub-tropical ridge over central Australia is thought to be the cause for the decrease in correlation strength from the north to the south of the NATT. During winter the change in the direction of this trend may be due to the typical northward shift of the sub-tropical ridge (Sturman and Tapper, 2006) and a relationship between SSTs off northwest Australia and the large northwest-southeast cloud bands (Wright, 1988a, 1988b), both of which are discussed more later in this section.

Of the ENSO indices, all showed significant correlations at most sites during spring, the strength of which ranged from weak to moderate for the Niño indices (maximum $r^2 = 0.19$) and weak to moderate, and weak to strong for EMI and SOI respectively (maximum $r^2 = 0.139$ and $0.283$ respectively). At most sites the SOI showed moderate to weak correlations during summer (maximum $r^2 = 0.18$) and mainly weak correlations during winter (maximum $r^2 = 0.11$), whereas for the SOI during autumn and the EMI and Niño indices for all seasons excluding spring, few significant correlations were evident.

AUSMI showed significant correlations over the whole NATT for all seasons, except in the north during winter. Correlations were highest in summer (strong in the north and moderate to weak in the south) and remained mostly moderate during autumn. Correlations during spring, while significant at all sites, were relatively weak. These results suggest AUSMI is best at describing rainfall at the peak and the end of the wet season, covering the majority of monsoonal rainfall, but is not necessarily good at describing the monsoon onset. AUSMI and II had some of the highest correlations in the south of the NATT during

winter (Table 4). Nicholls (1989) found that Australian winter rainfall was related to Indian Ocean and Indonesian SSTs and theorised that the SSTs over these regions have an influence on the location of the sub-tropical ridge along the east coast of Australia, which in turn influences rainfall. Wright (1988a) and Wright (1988b) found variability in southeastern Australian rainfall could be related to the presence of large cloud bands which form over the Indian Ocean, extending from northwest
Australia, across the south of the NT, to southeast Australia. These past studies could help to explain why SSTs off the north of Australia showed higher correlations with rainfall in the south of the NATT rather than with rainfall in the north.

EPI and DMI only exhibited significant correlations during spring, coinciding with the peak of the DMI (Saji et al., 1999). While WPI did not show significant correlations during any season it cannot be assumed to have no effect on NT rainfall since
it is a component of the DMI. The II showed significant correlations over the majority of the NATT during winter and spring only. IOD indices and the II are both calculated from SSTs near Indonesia and over the Indian Ocean. It is therefore possible that rainfall in winter and spring (but not summer or autumn) is influenced by climatic phenomena over the Indian Ocean and near Indonesia. Significant TSI-rainfall correlations were evident at almost all NATT sites during summer and spring as well as in the south during autumn. During autumn, TSI correlations were greatest in the south, while during summer they were
greatest in the north. The presence of stronger correlations in the north during summer may be due to the fact that monsoon onset can be triggered by extra-tropical weather events (Sturman and Tapper, 2006) which, as mentioned earlier in Section 3.3.2, may be linked to Tasman Sea SSTs (i.e. the TSI). Over all seasons the PDO, NATL, SATL and TROP showed almost no significant correlation with rainfall, except for SATL in the north during autumn. The lack of correlation between these indices and rainfall suggests there are probably no strong linear teleconnections between NT rainfall and North Pacific SSTs,
Atlantic Ocean SSTs, or zonal tropical SSTs on an annual timeframe.

This research has shown that correlations between NATT rainfall and climate indices are generally greater in the north. This is potentially due to the fact that, as discussed earlier, rainfall is more variable from year to year in the south. The dominant influence of the monsoon on rainfall in the north of the NATT, and the influence of various climatic phenomena on the
monsoon, also potentially explain why northern correlation strengths are strongest. Another cause for the higher rainfall-climate index correlation strengths in the north could be the presence of stronger trends in rainfall in the north. Correlations between rainfall and climate indices which also have significant trends could result in higher correlation strengths than would be expected if these data had been detrended before being analysed.

### 3.4 Implications

Given the strong correlations between rainfall and indices derived from the Indian-Australian monsoon, ENSO, El Niño-Southern Oscillation Modoki, IOD, and Tasman and Indonesian SSTs, there may be important atmosphere-ocean mechanisms

in the region that are crucial for climate dynamics and regional rainfall. Therefore climate models must be able to capture these dynamics in order to more accurately project future rainfall change in the NT.

### 3.4.1 Climate change

Over this study period there was a weakening in AUSMI, suggesting that the monsoon intensity may be weakening or that monsoon duration may be decreasing. Li et al. (2012) also noted a weakening of the monsoon from 1948 to 2007 despite increasing rainfall, therefore suggesting that the rainfall dynamics may be changing. Influences such as changes in atmospheric constituents (Rotstayn et al., 2007) and temperature (Wardle and Smith, 2004) may modulate rainfall variability. This research has not investigated whether the strength of the correlations between AUSMI and NATT rainfall have changed over time which may provide some insight into the likelihood of this theory, suggesting an avenue for further research.

Using modelling, Vecchi and Soden (2007) found that the Walker circulation is expected to weaken over the 21st century, with an associated shift to more El Niño-like conditions, and the IPCC's Fourth Assessment Report noted that ENSO-monsoon relationships will possibly experience a weakening in a warmer climate (Meehl et al., 2007). If the link between the monsoon and ENSO weakens, as suggested by Kumar et al. (1999), the usefulness of ENSO related indices in describing future Australian rainfall may be reduced. More research is required to determine what effect these changes may have on NT rainfall.

### 3.4.2 Implications for human populations and ecosystems

This research has reinforced the notion that future water availability is uncertain due to both unknown trends and relationships between climatic phenomena and natural climate variability, and the uncertainty surrounding the effect of anthropogenic climate change on precipitation. The net effect of climate change is expected to be detrimental to ecosystems (Yi et al., 2010) and biodiversity (IPCC, 2007b) thus affecting water availability and food production (Kanniah et al., 2010). Climate and land use change is expected to cause savannisation in some tropical forests, such as in the eastern Amazonia (IPCC, 2007b), further increasing the need to understand climatic processes and causes of rainfall in these biomes. The NATT and the entire OzFlux network (Beringer et al., 2016) provides an ideal platform to study changes in savanna vegetation composition, density and ecosystem energy balance over time due to the low intervention by humans (Beringer et al., 2011b). The strong rainfall gradient along the NATT has the potential to be used for an investigation of how variability in rainfall amount and reliability can affect savanna ecosystems. The use of a rainfall gradient could also be used to project how savanna ecosystems may change in the future due to potential rainfall changes associated with anthropogenic climate change.

**4 Conclusions**

This research has found that AUSMI had the strongest relationship with monthly rainfall. This finding is consistent with past research which shows that much of the change in rainfall and wet season characteristics from the north to the south of the

NATT is due to the decreasing influence of the monsoon southwards along the transect. These changes were linked to the decrease of the influence of the monsoon inland (Cook and Heerdegen, 2001), creating dramatic differences in ecosystem type, function and vegetation density along the transect (Beringer et al., 2011b). The dominance of the monsoon in the north of the NATT creates an environment where yearly summer rainfall is consistently high. In contrast, the southern NATT experienced more variability in rainfall amounts, which was most likely due to the varying influence of many different climatic phenomena.

The north of the NATT experienced less relative variability in wet season total, start and end date, duration, number of rain days and intensity than the south. It must be noted that AUSMI and NATT rainfall at all sites had significant linear trends which has potentially created correlation strengths that are unrealistically high.

This research has confirmed past findings that SOI is correlated with NT rainfall (e.g. Risbey et al., 2009; Schepen et al., 2012)

and has provided a better understanding of correlations between lesser used climate indices such as the TSI and the II and how their relationship with NATT rainfall changes over the course of a year. We suggest that rainfall over the NATT may have teleconnections with Indonesian, Tasman and Pacific SSTs and the climatic phenomena that affect these regions, however some of these relationships might be due to linear trends in the data. Past research has demonstrated that some climate phenomena-rainfall relationships do not remain constant over time (e.g. Nicholls et al., 1996; Power et al., 1999; Risbey et al.,

2009). This uncertainty is heightened due to the unknown influences of anthropogenic climate change on atmospheric processes and various climate phenomena-rainfall relationships (Murphy and Ribbe, 2004). Further work is needed to assess long term trends in atmosphere-ocean coupling and the potential affect of anthropogenic climate change on climatic phenomena.

This research has opened the door to many future studies concerning the use of climate indices to describe NT and savanna rainfall. This study has presented opportunities to expand this research to examine changes in climate index-rainfall correlation strength over time and the inclusion of more climatic phenomena.

**Acknowledgements**

This work was funded by the Australian Research Council (DP130101566). Beringer is funded under an ARC FT (FT1110602). Support for collection and archiving was provided through the Australia Terrestrial Ecosystem Research

Network (TERN) (http://www.tern.org.au). We would like to thank Darien Pardinas for programming and data processing, Neville Nicholls and Nigel Tapper for their constructive comments, Matt Paget and the CSIRO for access to the AWAP data, the NCDC: NOAA for access to the ERSST.v3b data, the Australian Bureau of Meteorology for access to the SOI data, the University of Washington: JISAO for access to PDO data, and the NOAA: ESRL for access to the 20CR data. Support for the

Twentieth Century Reanalysis Project dataset is provided by the U.S. Department of Energy, Office of Science Innovative and Novel Computational Impact on Theory and Experiment (DOE INCITE) program, and Office of Biological and Environmental Research (BER), and by the National Oceanic and Atmospheric Administration Climate Program Office.

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

**Table 1: Names, locations and mean annual precipitation (MAP) for sites along the North Australian Tropical Transect. Values in brackets show coefficients of variation. MAP from AWAP data from 1900 to 2010.**

| Site Name | Abbreviated Name | Latitude (°S) | Longitude (°E) | MAP (mm) |
|---|---|---|---|---|
| Howard Springs | HS | 12.49 | 131.15 | 1520 (19%) |
| Fogg Dam | FD | 12.55 | 131.31 | 1423 (20%) |
| Adelaide River | AR | 13.08 | 131.12 | 1318 (22%) |
| DDRF | DD | 13.83 | 131.19 | 1100 (22%) |
| Daly River Pasture (post cow) | DR | 14.06 | 131.32 | 1060 (21%) |
| Daly Uncleared | DU | 14.16 | 131.39 | 1041 (22%) |
| Dry Creek (a.k.a. Dry River) | DC | 15.26 | 132.37 | 762 (27%) |
| Sturt Plains | SP | 17.15 | 133.35 | 521 (37%) |
| Banka Banka Station | BB | 18.71 | 133.92 | 394 (45%) |
| Warumungu | Wa | 19.89 | 134.25 | 345 (47%) |
| Wycliffe Well | WW | 20.81 | 134.23 | 298 (50%) |
| Barrow Creek | BC | 21.50 | 133.92 | 294 (55%) |
| Alice Springs Flux Tower | AS | 22.28 | 133.25 | 288 (52%) |
| Tiger Bush | TB | 23.37 | 133.84 | 274 (55%) |
| MacDonalds | Ma | 24.47 | 133.47 | 225 (58%) |
| Ghan | Gh | 25.50 | 133.25 | 189 (62%) |

4    **Table 2: Climate index information showing index name, related climatic phenomenon, variable measured, region of influence, data source, and a key reference.**

| Climate Index | Climate Index Name | Climatic Phenomenon | Variable | Region | Data Source | References |
|---|---|---|---|---|---|---|
| WPI | Indian Ocean West Pole Index | IOD Component | SSTA | Indian Ocean | ERSST.v3b | (Saji et al., 1999) |
| EPI | Indian Ocean East Pole Index | IOD Component | SSTA | Indian Ocean | ERSST.v3b | (Saji et al., 1999) |
| DMI | Indian Ocean Dipole Mode Index | IOD | SSTA difference | Indian Ocean | ERSST.v3b | (Saji et al., 1999) |
| SOI | Southern Oscillation Index | ENSO | MSLP difference | Pacific Ocean | BoM | (Bureau of Meteorology, 2012) |
| TSI | Tasman Sea Index | Extratropical Climate | SSTA | Tasman Sea | ERSST.v3b | (Murphy and Timbal, 2008) |
| Nino1+2 | Extreme Eastern Tropical Pacific SST | ENSO | SSTA | Pacific Ocean | ERSST.v3b | (ESRL, 2012) |
| Nino3 | Eastern Tropical Pacific SST | ENSO | SSTA | Pacific Ocean | ERSST.v3b | (ESRL, 2012) |
| Nino3.4 | East Central Tropical Pacific SST | ENSO | SSTA | Pacific Ocean | ERSST.v3b | (ESRL, 2012) |
| Nino4 | Central Tropical Pacific SST | ENSO | SSTA | Pacific Ocean | ERSST.v3b | (ESRL, 2012) |
| NATL | North Atlantic Index | Atlantic Ocean | SSTA | Atlantic Ocean | ERSST.v3b | (National Oceanic and Atmospheric Administration, 2012) |
| SATL | South Atlantic Index | Atlantic Ocean | SSTA | Atlantic Ocean | ERSST.v3b | (National Oceanic and Atmospheric Administration, 2012) |
| TROP | Global Tropics Index | Tropical Oceans | SSTA | Equatorial | ERSST.v3b | (National Oceanic and Atmospheric Administration, 2012) |
| EMI | ENSO Modoki Index | ENSO | SSTA calculation | Pacific Ocean | ERSST.v3b | (Ashok et al., 2007) |
| II | Indonesia Index | Indonesia SST | SSTA | Indonesia | ERSST.v3b | (Verdon and Franks, 2005) |
| AUSMI | Australian Monsoon Index | Indian-Australian Monsoon | 850mb U-wind | Indonesia | 20CR | (Kajikawa et al., 2009) |
| PDO | Pacific Decadal Oscillation | PDO | SST PC | Pacific Ocean | JISAO | (Joint Institute for the Study of the Atmosphere and Ocean, 2012) |

**Table 3: Temporal rainfall trends using linear regression of annual data along the NATT for the period 1900 to 2010. Italicised and bold $r^2$ values are significant at the 95% significance level (i.e. all trends are significant). Strength shows the goodness of fit of the linear trend line. Absolute gradient shows the actual increase in annual rainfall per decade. Fractional gradient shows the increase in annual rainfall per decade of a site relative to the mean rainfall at that site. Brackets indicate the sign of the linear trend if significant. Coefficient of variation shows the variability of each site as a percentage of the sites annual average rainfall.**

| Site | Latitude (°S) | Correlation Coefficient ($r^2$) | Absolute Gradient (mm decade$^{-1}$) | Fractional Gradient (% change per decade) | Coefficient of Variation |
|------|------|------|------|------|------|
| HS | 12.49 | *0.14* | 33.20 (+) | 2.18 (+) | 19% |
| FD | 12.55 | *0.17* | 36.52 (+) | 2.57 (+) | 20% |
| AR | 13.08 | *0.22* | 41.71 (+) | 3.16 (+) | 22% |
| DD | 13.83 | *0.16* | 29.82 (+) | 2.71 (+) | 22% |
| DR | 14.06 | *0.17* | 28.52 (+) | 2.69 (+) | 21% |
| DU | 14.16 | *0.19* | 30.25 (+) | 2.91 (+) | 22% |
| DC | 15.26 | *0.22* | 30.54 (+) | 4.01 (+) | 27% |
| SP | 17.15 | *0.13* | 21.77 (+) | 4.18 (+) | 37% |
| BB | 18.71 | *0.04* | 10.78 (+) | 2.74 (+) | 45% |
| Wa | 19.89 | *0.06* | 12.60 (+) | 3.65 (+) | 47% |
| WW | 20.81 | *0.11* | 14.99 (+) | 5.02 (+) | 50% |
| BC | 21.50 | *0.10* | 15.76 (+) | 5.35 (+) | 55% |
| AS | 22.28 | *0.10* | 14.55 (+) | 5.06 (+) | 52% |
| TB | 23.37 | *0.06* | 11.10 (+) | 4.05 (+) | 55% |
| Ma | 24.47 | *0.06* | 9.70 (+) | 4.32 (+) | 58% |
| Gh | 25.50 | *0.06* | 8.64 (+) | 4.57 (+) | 62% |

13 **Table 4: Table shows the first three most highly correlated indicies with rainfall at each site over the NATT (Table 1) for the period 1900 to 2010. Correlations are shown using**
14 **the monthly data and inter-annual correlations are provided using the annual data. In addition, annual correlations are computed for the individual seasons only. Only**
15 **correlations significant at the 95% level are shown. Colours represent in which region the climate index is located and are defined as follows; red = Indonesia and Indian Ocean,**
16 **light blue = Pacific Ocean, dark blue = Tasman Sea, orange = Atlantic and tropical ocean. The first column for each time period in this table (i.e. the "1" columns) are shown in**
17 **Figure 5.**

| Site | Lat (°S) | Monthly | | | Annual | | | Summer | | | Autumn | | | Winter | | | Spring | | |
|---|---|---|---|---|---|---|---|---|---|---|---|---|---|---|---|---|---|---|---|
| | | 1 | 2 | 3 | 1 | 2 | 3 | 1 | 2 | 3 | 1 | 2 | 3 | 1 | 2 | 3 | 1 | 2 | 3 |
| HS | 12.5 | AUSMI | SOI | WPI | SOI | TSI | II | AUSMI | SOI | TSI | AUSMI | EMI | SOI | SOI | - | - | SOI | Nino3.4 | Nino3 |
| FD | 12.6 | AUSMI | SOI | WPI | TSI | II | SOI | AUSMI | SOI | TSI | AUSMI | SATL | SOI | SOI | - | - | SOI | II | Nino3.4 |
| AR | 13.1 | AUSMI | SOI | WPI | TSI | II | SOI | AUSMI | TSI | SOI | AUSMI | EMI | SATL | SOI | II | - | II | SOI | EPI |
| DD | 13.8 | AUSMI | SOI | DMI | TSI | EMI | II | AUSMI | TSI | SOI | AUSMI | EMI | SOI | SOI | II | AUSMI | II | EPI | SOI |
| DR | 14.1 | AUSMI | SOI | DMI | TSI | II | EMI | AUSMI | SOI | TSI | AUSMI | EMI | SOI | SOI | - | - | II | EPI | SOI |
| DU | 14.2 | AUSMI | SOI | WPI | TSI | II | EMI | AUSMI | TSI | SOI | AUSMI | EMI | SOI | - | - | - | II | SOI | EPI |
| DC | 15.3 | AUSMI | SOI | WPI | II | TSI | SATL | AUSMI | SOI | TSI | AUSMI | SATL | SOI | SOI | II | - | SOI | II | DMI |
| SP | 17.2 | AUSMI | SOI | WPI | TSI | SOI | II | AUSMI | SOI | TSI | AUSMI | SOI | - | SOI | II | Nino3.4 | SOI | II | TSI |
| BB | 18.7 | AUSMI | SOI | DMI | TSI | AUSMI | SOI | AUSMI | SOI | TSI | AUSMI | - | - | SOI | AUSMI | II | SOI | DMI | Nino3.4 |
| Wa | 19.9 | AUSMI | SOI | WPI | AUSMI | TSI | - | AUSMI | SOI | TSI | AUSMI | - | - | SOI | AUSMI | SATL | SOI | II | EPI |
| WW | 20.8 | AUSMI | SOI | TSI | TSI | II | AUSMI | AUSMI | SOI | DMI | TSI | AUSMI | - | SOI | II | AUSMI | II | SOI | TSI |
| BC | 21.5 | AUSMI | SOI | TSI | TSI | II | AUSMI | AUSMI | TSI | DMI | AUSMI | - | - | AUSMI | SOI | WPI | II | SOI | AUSMI |
| AS | 22.3 | AUSMI | SOI | TSI | TSI | II | AUSMI | TSI | AUSMI | SATL | AUSMI | TSI | - | AUSMI | WPI | SOI | II | EPI | SOI |
| TB | 23.4 | AUSMI | SOI | TSI | TSI | AUSMI | SOI | AUSMI | SOI | TSI | AUSMI | TSI | - | AUSMI | II | SOI | SOI | II | TSI |
| Ma | 24.5 | AUSMI | SOI | TSI | TSI | AUSMI | SATL | AUSMI | SOI | - | AUSMI | TSI | - | AUSMI | II | SOI | SOI | II | AUSMI |
| Gh | 25.5 | AUSMI | SOI | II | TSI | II | EMI | AUSMI | - | - | TSI | AUSMI | II | II | DMI | AUSMI | II | SOI | EPI |

**Table 5: Ranks showing relative correlation strengths of each climate index with precipitation over monthly, annual and seasonal time periods. Each rank shows, on average,**
**which climate index has the highest correlation with rainfall at all sites along the NATT from July 1900 to June 2010 (except winter which uses calendar year from 1900 to 2009,**
**not hydrological year). For example, on a monthly time scale AUSMI, on average, is the most highly correlated climate index over the entire NATT, SOI is the second most**
**highly correlated, EMI is the third highest, and so on. Note, this table does not take statistical significance into account. First\*\*\*, second\*\* and third\* highest correlations are**
**shaded and given a number of asterisk's for clarity.**

| Rank | Climate Index | | | | | | | | | | | | | | | |
|---|---|---|---|---|---|---|---|---|---|---|---|---|---|---|---|---|
| | WPI | EPI | DMI | SOI | TSI | Niño 1+2 | Niño3 | Niño 3.4 | Niño4 | NATL | SATL | TROP | EMI | II | AUSMI | PDO |
| Monthly | 4 | 14.5 | 7 | 2** | 5 | 16 | 12 | 8 | 11 | 13 | 10 | 14.5 | 3* | 6 | 1*** | 9 |
| Annual | 10 | 7 | 13 | 3* | 1*** | 16 | 11 | 8 | 12 | 14 | 6 | 15 | 4 | 2** | 5 | 9 |
| Summer | 9 | 15 | 7 | 2** | 3* | 14 | 8 | 6 | 13 | 10 | 4 | 16 | 11 | 12 | 1*** | 5 |
| Autumn | 13 | 9 | 8 | 2** | 4 | 11.5 | 14 | 10 | 6 | 11.5 | 5 | 15 | 3* | 7 | 1*** | 16 |
| Winter | 8 | 16 | 7 | 1*** | 15 | 6 | 11 | 4.5 | 4.5 | 9 | 13 | 14 | 10 | 2.5** | 2.5** | 12 |
| Spring | 15 | 3* | 4 | 1*** | 6 | 11 | 9 | 7 | 10 | 16 | 12.5 | 14 | 5 | 2** | 8 | 12.5 |

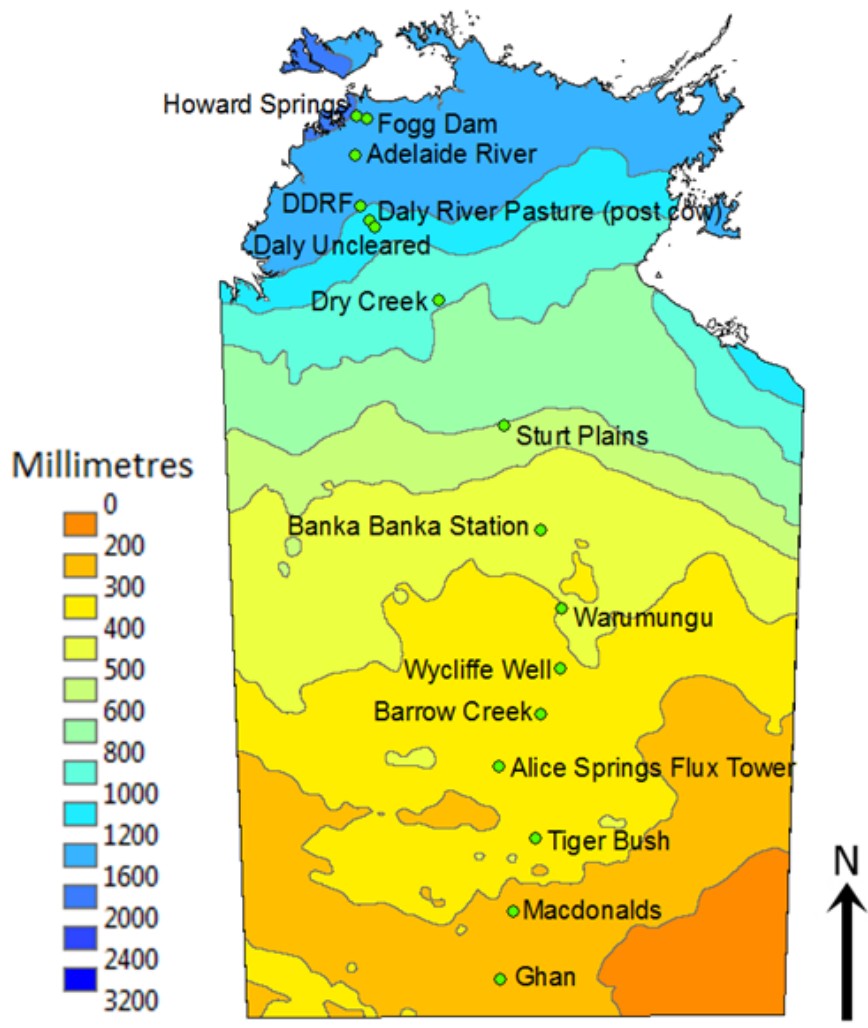

**Figure 1: Map of average annual rainfall (mm) from 1961 to 1990 (colours) and the location of each site along the North Australian Tropical Transect (Adapted from Bureau of Meteorology (2011a)).**

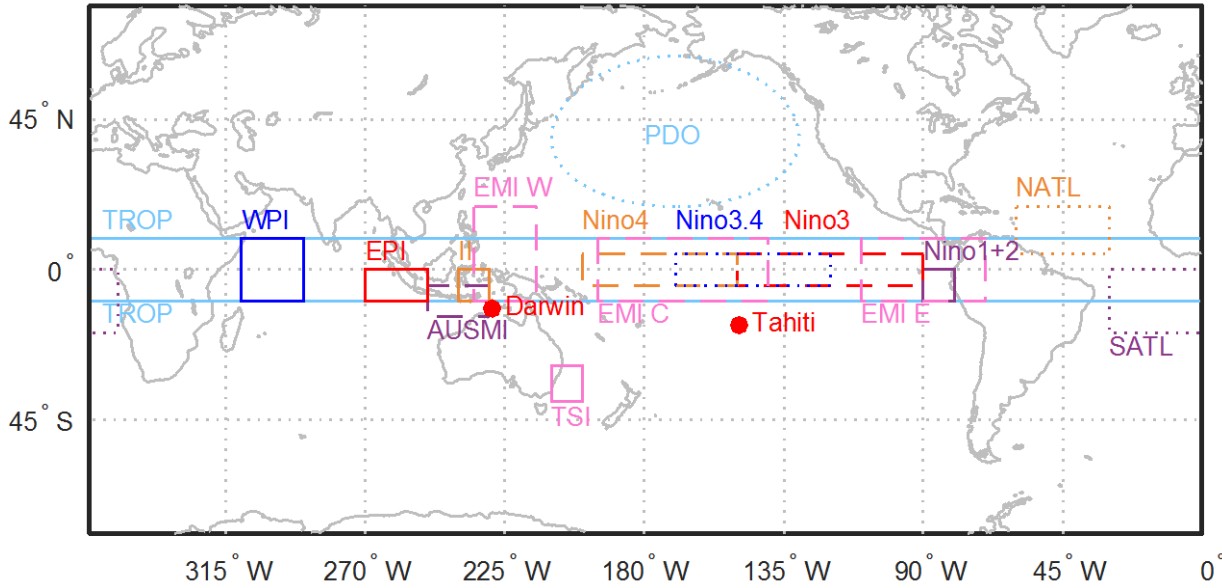

**Figure 2: Approximate climate index measurement regions. Climate indices are calculated over the following areas: WPI – solid blue rectangle, EPI – solid red rectangle, DMI – the difference between the WPI and the EPI, SOI – calculated using MSLP measurements at Tahiti and Darwin (red dots), TSI – solid pink rectangle, Niño 1+2 – solid purple rectangle, Niño 3 – dashed red rectangle, Niño 3.4 – dotted blue rectangle, Niño 4 – dashed orange rectangle, NATL – dotted orange rectangle, SATL – dotted purple rectangle, TROP – calculated between top and bottom light blue lines, EMI – calculated using areas enclosed by dashed pink rectangles, II – solid orange rectangle, AUSMI – dashed purple rectangle and PDO – dashed light blue oval (note: PDO is calculated over the entire area of the Pacific Ocean north of 20°N, the oval is only included for illustrative purposes). Figure produced using definitions for climate indices listed in Section 2.2.**

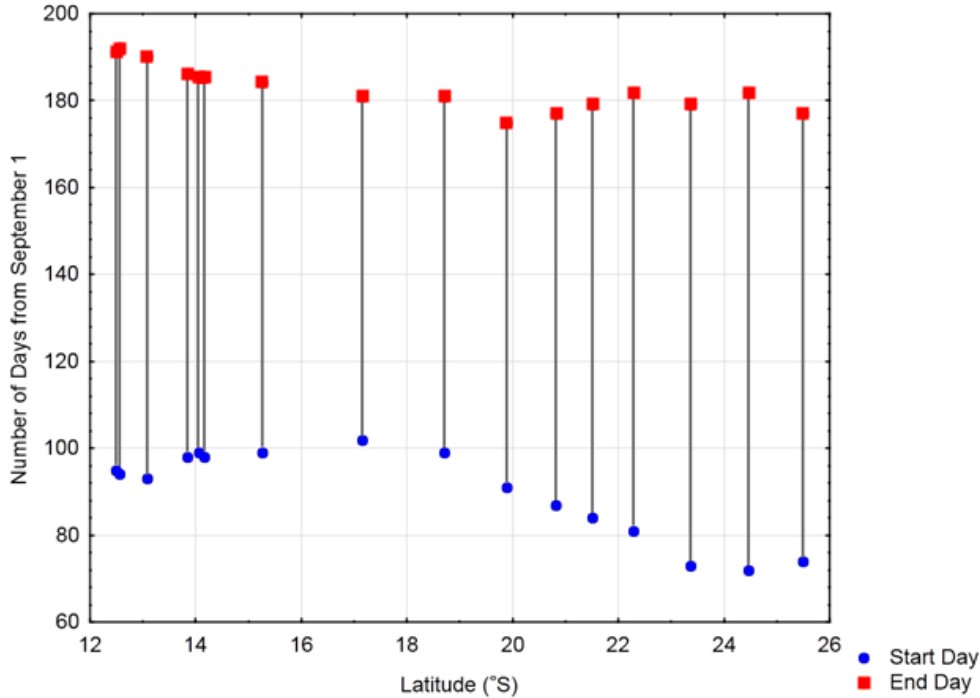

**Figure 3: Wet season characteristics for each site along the NATT (Table 1) for the period 1900 to 2010. Sites are plotted as their latitude along the transect. Average wet season start day, end day and duration are given. Start and end day are defined as the date when rainfall, between September 1 and April 30 the following year, exceeded 15% and 85% respectively of the total rainfall between these two dates (Smith et al., 2008a). Start and end days are displayed as the number of days from September 1. Wet season duration is shown by the grey line connecting start and end days.**

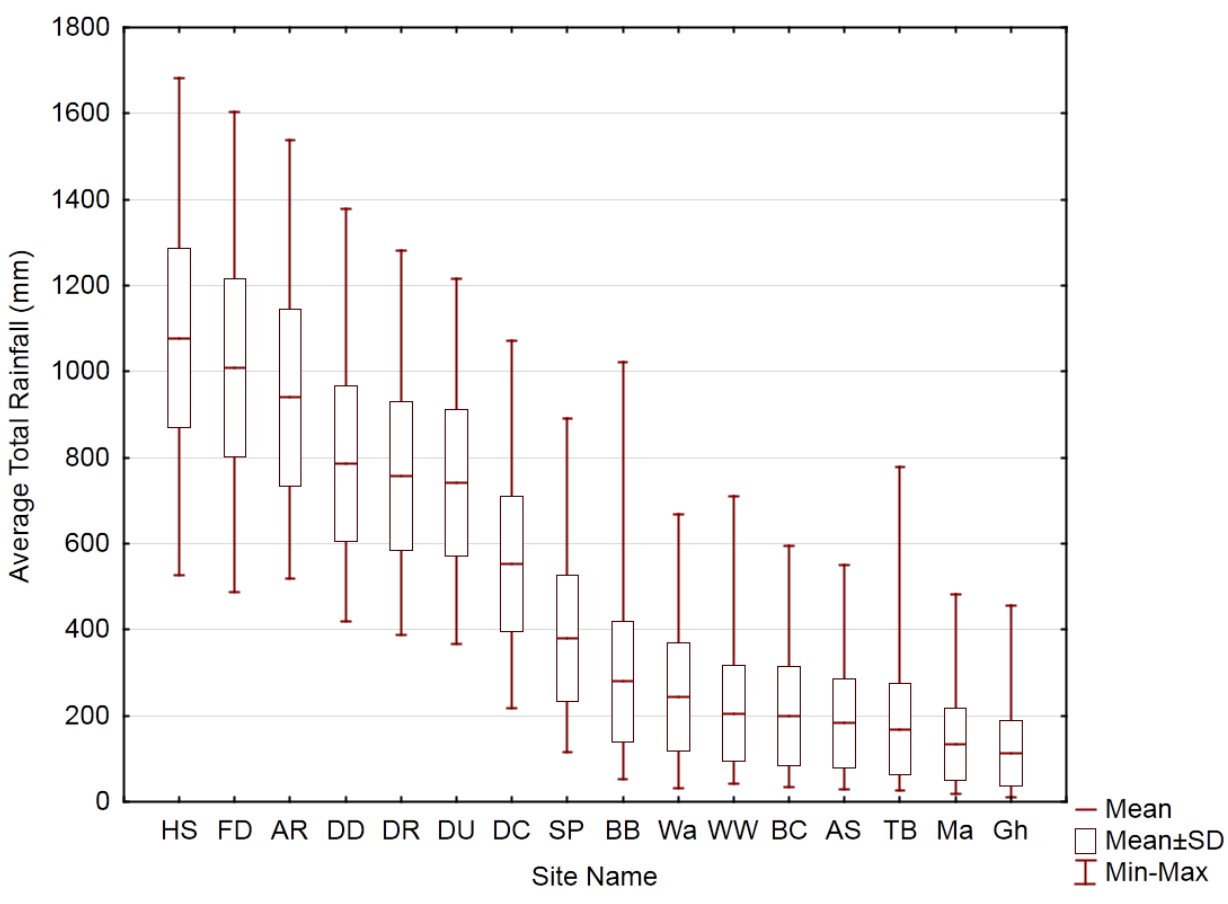

**Figure 4: Mean annual wet season rainfall (mm) from 1900 to 2010 for each site along the NATT (Table 1). Boxes show one standard deviation and whiskers show maximum and minimum seasonal rainfall for each site.**

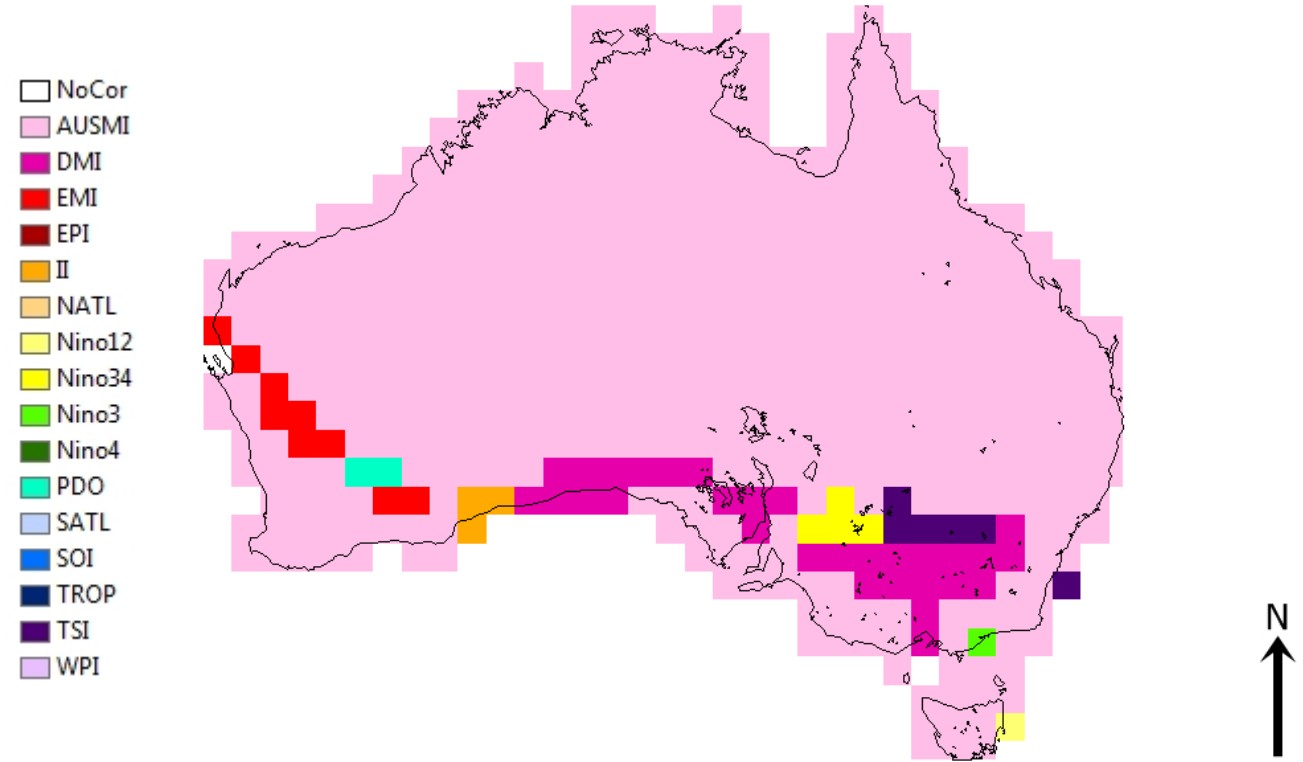

**Figure 5: Map showing the climate index that correlates most strongly with monthly total rainfall over Australia between 1900 and 2010. Data aggregated to a resolution of 1.25° by 1.25° (approximately 125 km by 125 km). All coloured grid cells show correlations that are significant at the 95% significance level.**

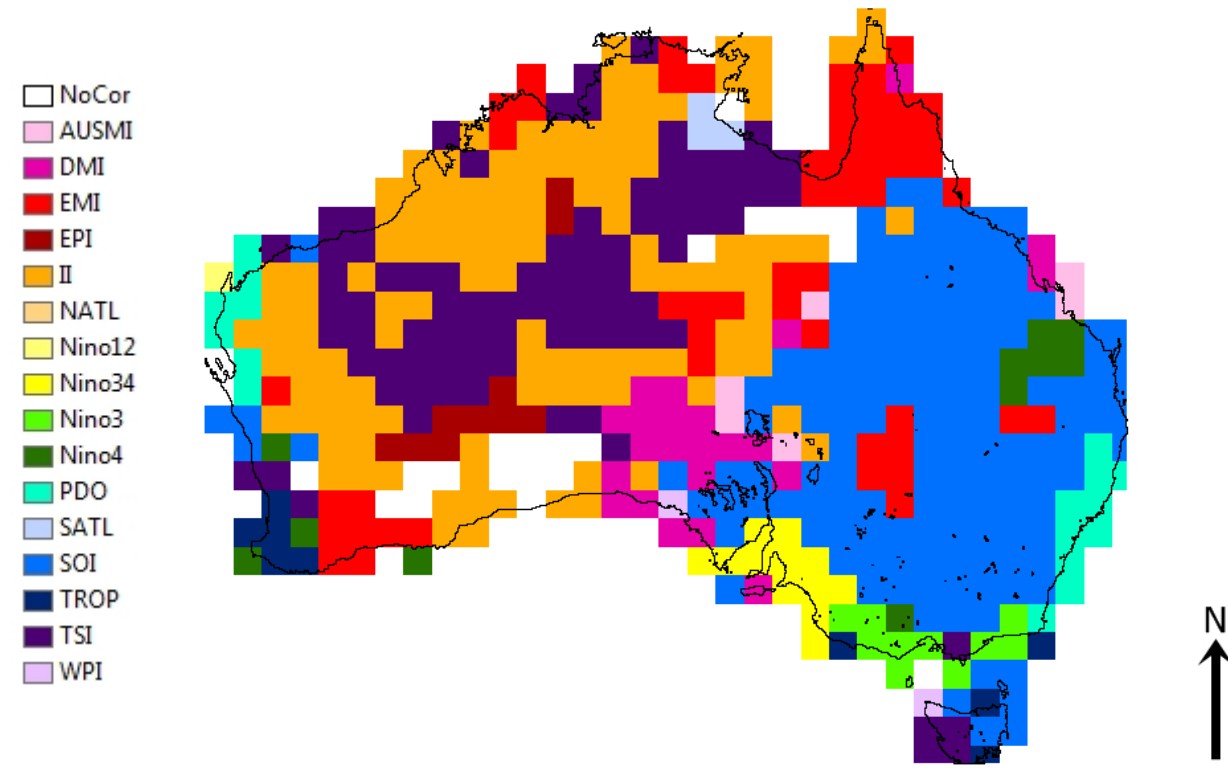

**Figure 6: Map showing the climate index that correlates most strongly with annual total rainfall over Australia between 1900 and 2010. Data aggregated to a resolution of 1.25° by 1.25° (approximately 125 km by 125 km). All coloured grid cells show correlations that are significant at the 95% significance level. White grid cells show areas with no significant correlations.**

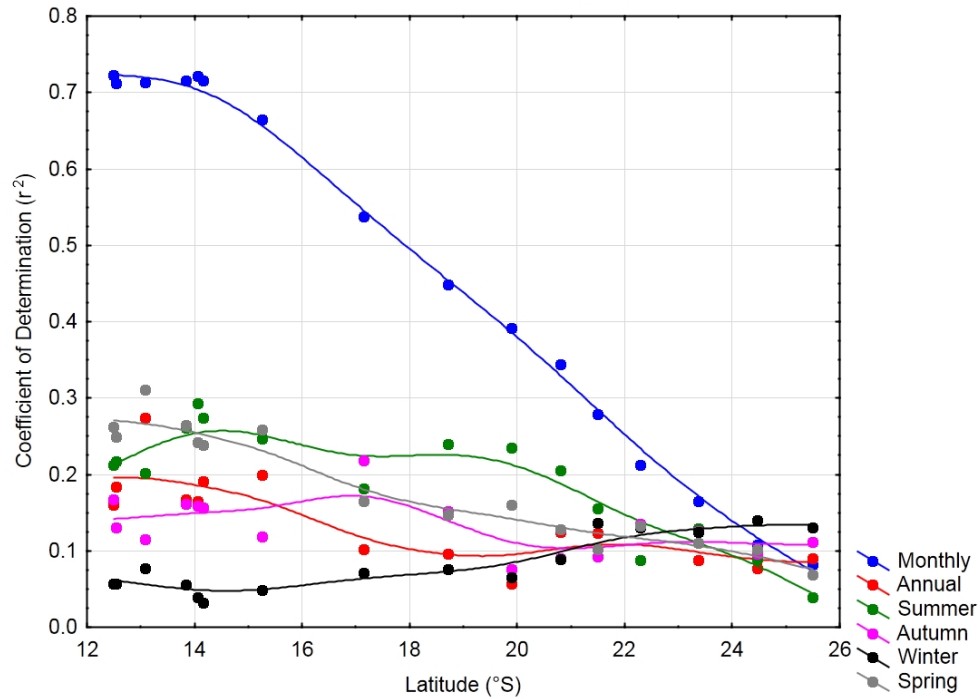

**Figure 7: Correlations between rainfall and the climate index of maximum correlation at each site along the NATT (Table 1) for the period 1900 to 2010. Sites are plotted as their latitude along the transect. A distance weighted least squares fit has been included to enhance the visualisation of the trends along the NATT. The climate indices used for this graph vary depending on site and time period. Indices used for each point on the graph can be found in the "1" columns (highest correlation columns) of Table 4.**

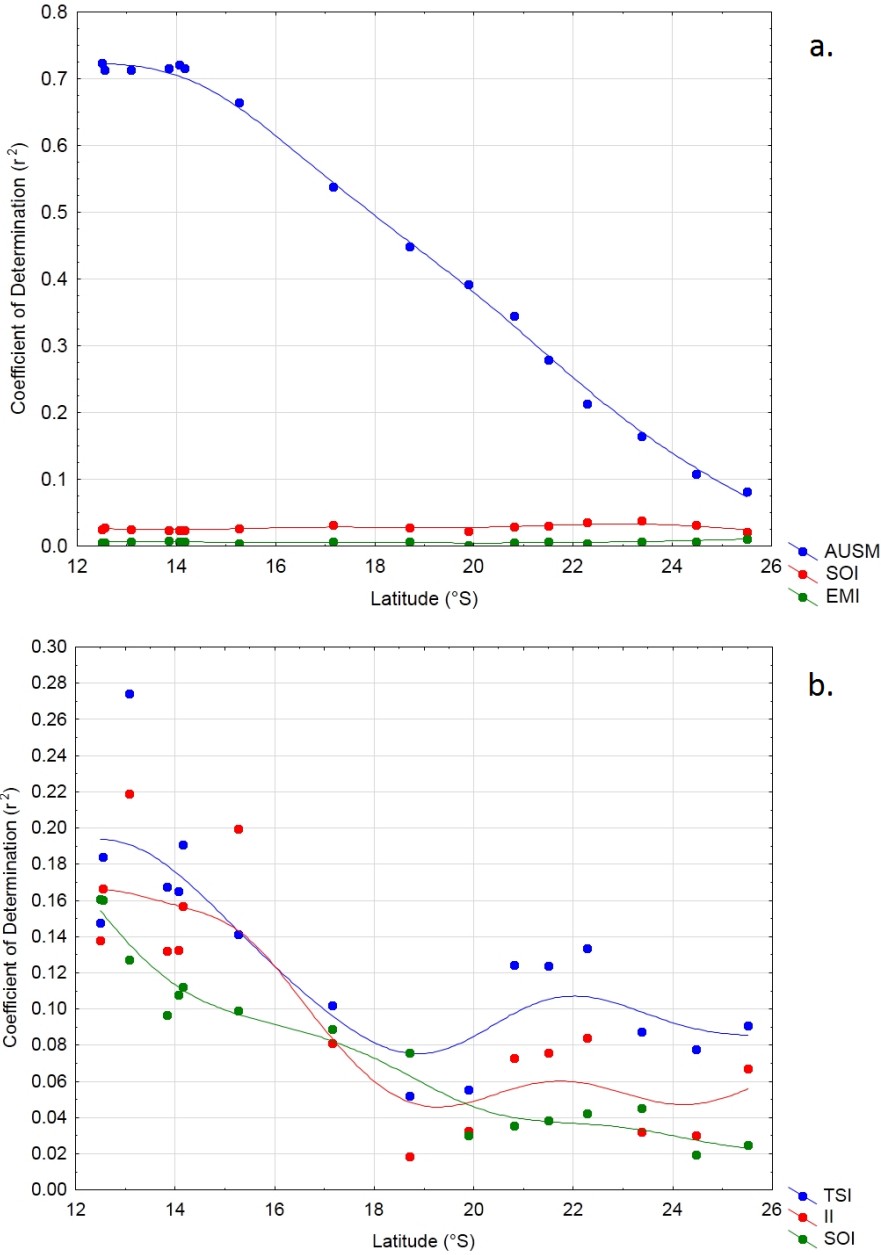

**Figure 8: The three most highly correlated climate indices (r²) on average along the NATT (Table 1) for the period 1900 to 2010. Sites are plotted as their latitude along the transect. Correlations are shown for (a) monthly and (b) yearly averaged daily rainfall. For monthly correlations, AUSMI had the strongest correlation, followed by SOI, then EMI. For annual correlations, TSI showed the strongest correlation on average over the entire NATT, followed by II then SOI. A distance weighted least squares fit has been included to aid the illustration of the trends along the NATT.**

