# Peer review of "Describing rainfall in northern Australia using multiple climate indices"

_Biogeosciences, 2016_

## Referee Comment (RC1) · Anonymous Referee #1 · 19 May 2016

General comments This manuscript describes an interesting study designed to evaluate how well several climate indices correlated with the spatial and temporal patterns of rainfall along a severe rainfall gradient in the Northern Territory, Australia. The investigators used a rainfall record from 1900 to 2010 and correlated the climate indices to rainfall at 16 locations along a rainfall gradient from 1600 mm/y to 200 mm/y. They investigated the relationship at annual, seasonal and monthly scales. The sites used are known at the North Australian Tropical Transect (NATT). The study found that across the NATT the AUSMI index provided the best correlations at a monthly time step while the TSI index was the best predictor at an annual time step. The study only examined correlations, so no cause and effect relationships could be determined.

Specific comments Recommendations for improvements to the manuscript: 1. Revise the Introduction to focus more on the topic of this study rather than on the ancillary

topics of climate change and vegetation dynamics. Of course, variability in rainfall can have important effects on vegetation and is likely to change under future climates, but the study did not examine those things.

2. The presentation of the results was generally clear, but the Australia-wide data presented in Figures 5 and 6 and Tables 4 and 5 seemed out of place. I recommend putting that information in a supplementary materials section, or perhaps at the end of the Results and Discussion section as a separate topic, or omitting it.

3. Similar to (2), if I correctly understood the time lag analyses mentioned briefly in sections 2.5 and 3.3.4, they were conducted at continental and Northern Territory scales and also do not fit well with the other analyses done using the NATT. I recommend either omitting the discussion of the time lag analysis or more fully incorporating it into the manuscript by providing additional information and data.

4. I think the Results and Discussion section would be improved by discussing the strength of the correlations between the climate indices and rainfall across the NATT. The authors pointed out that the correlations were highest at the northern end of the transect due to the dependability of the monsoon rainfall in that area, but did not address the low r2 of even the best-correlated index at the southern end of the transect, or that most of the climate indices were very poorly correlated with rainfall at all time scales across the entire NATT (see Figures 7 and 8). Discussing the reasons for this would be a useful addition to the manuscript.

Technical corrections p. 2, line 4, substitute effects for implications p. 3, line 19, "has been shown to feedback to affect..." is awkward p. 5, line 14 substitute in for is p. 5, line 20, mention where the rate change takes place along the transect p. 15, lines 13-14, mention that all of the correlations are low at the southern end of the transect p. 17, lines 23-24, this seems contradictory to Table 3 that shows an increase in rainfall over time for all points on the transect. p. 17, lines 25-29. This discussion is off topic. Please modify or omit. p. 18, lines 5-18. The first two paragraphs of the Conclusions

did not describe much about the key findings of the study. Either revise or omit.

---

## Referee Comment (RC2) · J. Hatfield (Referee) · 11 Aug 2016

This paper presents a comprehensive analysis of the rainfall distribution across northern Australia using different indices. It utilizes a network of information within the country and larger scale indices. The analysis is very robust and insightful.

---

## Author Comment (AC1)

**Response to reviewers comments on "Describing rainfall in northern Australia using multiple climate indices" by Cassandra Rogers and Jason Beringer**

We thank the reviewers for their thoughtful comments and support for the quality of the manuscript. We have taken the reviewers comments seriously and have used them to suggest improvements that will strengthen the manuscript.

**Anonymous Referee #1**

Received and published: 19 May 2016

**General comments**

This manuscript describes an interesting study designed to evaluate how well several climate indices correlated with the spatial and temporal patterns of rainfall along a severe rainfall gradient in the Northern Territory, Australia. The investigators used a rainfall record from 1900 to 2010 and correlated the climate indices to rainfall at 16 locations along a rainfall gradient from 1600 mm/y to 200 mm/y. They investigated the relationship at annual, seasonal and monthly scales. The sites used are known at the North Australian Tropical Transect (NATT). The study found that across the NATT the AUSMI index provided the best correlations at a monthly time step while the TSI index was the best predictor at an annual time step. The study only examined correlations, so no cause and effect relationships could be determined.

Specific comments Recommendations for improvements to the manuscript:

- 1. Revise the Introduction to focus more on the topic of this study rather than on the ancillary C1 topics of climate change and vegetation dynamics. Of course, variability in rainfall can have important effects on vegetation and is likely to change under future climates, but the study did not examine those things.
  - a. The reviewer raises a good point and the introduction will be revised to focus on rainfall variability and climate modes affecting the region.
- 2. The presentation of the results was generally clear, but the Australia-wide data presented in Figures 5 and 6 and Tables 4 and 5 seemed out of place. I recommend putting that information in a supplementary materials section, or perhaps at the end of the Results and Discussion section as a separate topic, or omitting it.
  - a. As suggested by the reviewer, the discussion referring to Figures 5 and 6 will be moved to a new section towards the end of the Results and Discussion section. These figures will be used to discuss the correlation strength between rainfall and climate indices over the NATT in the broader Australian context.
  - b. We argue that Table 4 is necessary for showing correlation strengths over the NATT and therefore should remain in the paper. Correlation strengths will be added to this Table to make it more useful.
  - c. Table 5 will be removed as recommended.
- Similar to (2), if I correctly understood the time lag analyses mentioned briefly in sections 2.5 and
  3.3.4, they were conducted at continental and Northern Territory scales and also do not fit well with the other analyses done using the NATT. I recommend either omitting the discussion of the time lag

analysis or more fully incorporating it into the manuscript by providing additional information and data.

- a. Additional information, including the addition of a figure, will be added to this section to better incorporate the time lag analysis
- 4. I think the Results and Discussion section would be improved by discussing the strength of the correlations between the climate indices and rainfall across the NATT. The authors pointed out that the correlations were highest at the northern end of the transect due to the dependability of the monsoon rainfall in that area, but did not address the low r2 of even the best-correlated index at the southern end of the transect, or that most of the climate indices were very poorly correlated with rainfall at all time scales across the entire NATT (see Figures 7 and 8). Discussing the reasons for this would be a useful addition to the manuscript.
  - a. More discussion will be added to address the nature of the correlations and physical processes that may be influencing these.

**Technical corrections**

- p. 2, line 4, substitute effects for implications
  - o Will change
- p. 3, line 19, "has been shown to feedback to affect. . ." is awkward
  - Will remove "to feedback"
- p. 5, line 14 substitute in for is
  - Changing in to is, i.e. from "which results in greenhouse gas emissions" to "which results is greenhouse gas emissions" sounds awkward. This change has not been made
- p. 5, line 20, mention where the rate change takes place along the transect
  - o Will mention
- p. 15, lines 13-14, mention that all of the correlations are low at the southern end of the transect
  - o Will mention
- p. 17, lines 23-24, this seems contradictory to Table 3 that shows an increase in rainfall over time for all points on the transect.
  - The section that the reviewer refers to states "This research has reinforced the notion that future water availability is uncertain due to both unknown trends and relationships between climatic phenomena and natural climate variability, and the uncertainty surrounding the effect of anthropogenic climate change on precipitation" and we feel there is nothing in this paragraph related to "increase in rainfall over time for all points on the transect" and therefore we have not modified this sentence.
- p. 17, lines 25-29. This discussion is off topic. Please modify or omit.
  - We do not agree that this section of the paper is off topic as it discusses some of the possible causes of the correlations between rainfall and the TSI. These sentences have been rewritten slightly to make this clearer.
- p. 18, lines 5-18.
  - We are not sure what the reviewers concern is with this section of the paper.

- The first two paragraphs of the Conclusions C2 did not describe much about the key findings of the study. Either revise or omit.
  - Will remove.

---

## Author Response (AR1)

**Response to reviewers comments on "Describing rainfall in northern Australia using multiple climate indices" by Cassandra Rogers and Jason Beringer**

We thank the reviewers for their thoughtful comments and support for the quality of the manuscript. We have taken the reviewers comments seriously and have used them to suggest improvements that will strengthen the manuscript. We have taken the opportunity to make additional small changes to improve the clarity throughout as well as redraw figure 2. We think this is a much improved manuscript.

Anonymous Referee #1

General comments

This manuscript describes an interesting study designed to evaluate how well several climate indices correlated with the spatial and temporal patterns of rainfall along a severe rainfall gradient in the Northern Territory, Australia. The investigators used a rainfall record from 1900 to 2010 and correlated the climate indices to rainfall at 16 locations along a rainfall gradient from 1600 mm/y to 200 mm/y. They investigated the relationship at annual, seasonal and monthly scales. The sites used are known at the North Australian Tropical Transect (NATT). The study found that across the NATT the AUSMI index provided the best correlations at a monthly time step while the TSI index was the best predictor at an annual time step. The study only examined correlations, so no cause and effect relationships could be determined.

Specific comments Recommendations for improvements to the manuscript:

1. Revise the Introduction to focus more on the topic of this study rather than on the ancillary topics of climate change and vegetation dynamics. Of course, variability in rainfall can have important effects on vegetation and is likely to change under future climates, but the study did not examine those things.

   a. The reviewer raises a good point and we have revised the introduction to focus on rainfall variability and climate modes affecting the region by removing some ancillary material.

2. The presentation of the results was generally clear, but the Australia-wide data presented in Figures 5 and 6 and Tables 4 and 5 seemed out of place. I recommend putting that information in a supplementary materials section, or perhaps at the end of the Results and Discussion section as a separate topic, or omitting it.

a. As suggested by the reviewer, the discussion referring to Figures 5 and 6 have been moved to a new section towards the end of the Results and Discussion section. These figures are used to discuss the correlation strength between rainfall and climate indices over the NATT in the broader Australian context.

b. We argue that Table 4 is necessary for showing correlation strengths over the NATT and therefore should remain in the paper. Correlation strengths will be added to this Table to make it more useful.

c. Table 5 has been removed as recommended.

3. Similar to (2), if I correctly understood the time lag analyses mentioned briefly in sections 2.5 and 3.3.4, they were conducted at continental and Northern Territory scales and also do not fit well with the other analyses done using the NATT. I recommend either omitting the discussion of the time lag analysis or more fully incorporating it into the manuscript by providing additional information and data.

a. We agree with the reviewer and have remove the time lag analysis that now streamlines the paper better.

4. I think the Results and Discussion section would be improved by discussing the strength of the correlations between the climate indices and rainfall across the NATT. The authors pointed out that the correlations were highest at the northern end of the transect due to the dependability of the monsoon rainfall in that area, but did not address the low $r^2$ of even the best-correlated index at the southern end of the transect, or that most of the climate indices were very poorly correlated with rainfall at all time scales across the entire NATT (see Figures 7 and 8). Discussing the reasons for this would be a useful addition to the manuscript.

a. More discussion has been added to address the nature of the correlations and physical processes that may be influencing these.

Technical corrections

- p. 2, line 4, substitute effects for implications
  - Have changed
- p. 3, line 19, "has been shown to feedback to affect. . ." is awkward
  - Have removed "to feedback"
- p. 5, line 14 substitute in for is
  - Changed in to is, i.e. from "which results in greenhouse gas emissions" to "which results is greenhouse gas emissions" sounds awkward.
- p. 5, line 20, mention where the rate change takes place along the transect
  - Have added "A sharp change in rainfall rate occurs around Dry Creek, with the stations north of Dry Creek experiencing higher rainfall, and the stations further south experiencing less rainfall."

- p. 15, lines 13-14, mention that all of the correlations are low at the southern end of the transect
  - Have added "Correlation coefficients between AUSMI and rainfall were very strong in the north of the transect, but weak in the south."
- p. 17, lines 23-24, this seems contradictory to Table 3 that shows an increase in rainfall over time for all points on the transect.
  - The section that the reviewer refers to states "This research has reinforced the notion that future water availability is uncertain due to both unknown trends and relationships between climatic phenomena and natural climate variability, and the uncertainty surrounding the effect of anthropogenic climate change on precipitation" and we feel there is nothing in this paragraph related to "increase in rainfall over time for all points on the transect" and therefore we have not modified this sentence.
- p. 17, lines 25-29. This discussion is off topic. Please modify or omit.
  - We do not agree that this section of the paper is off topic as it discusses some of the possible implications for humans and ecosystems. We think this is entirely appropriate for Biogeosciences that focuses on the intersection of disciplines. If this were a specialist climate journal then I would agree. In addition, this paper is a part of the OzFlux SI and creates some good links to the SI themes.
- p. 18, lines 5-18.
  - We are not sure what the reviewers concern is with this section of the paper.
- The first two paragraphs of the Conclusions did not describe much about the key findings of the study. Either revise or omit.
  - We have removed and it has streamlined the conclusions.

**Describing rainfall in northern Australia using multiple climate indices**

Cassandra Denise Wilks Rogers[1], Jason Beringer[1,2]

[revised manuscript text omitted]

5  In Australia, rainfall variability is crucial to the structure and productivity of the landscape, particularly across the vast extent of the savanna biome which accounts for (25% of the continent, (Beringer et al., (2007). Beringer et al., 2011b; Eamus et al., 2013; Haverd et al., 2016; Hutley et al., 2011; Kanniah et al., 2011). Not only do savanna landscapes cover a large portion of Australia, they account for 15% of the land surface of the planet  (Beringer et al., (2011b).) This area  has the potential to increase in size due to climate change (Franchito et al., 2011), making the
10  improved knowledge of these landscapes important in understanding future rainfall trends in Australia and around the world. Australian savannas remain fairly undisturbed (Beringer et al., 2014) making them good for temporal change studies. Vegetation productivity, (and hence the carbon balance,) are vulnerable to changes in rainfall variability (Kanniah et al., 2011) because savanna structure, composition and function shift in response to short (monsoonal) and long term (ENSO, Inter-
15  decadal Pacific Oscillation, Pacific Decadal Oscillation Index (PDO), etc.) rainfall climatology (Beringer et al., 2011a). Moreover, disturbances, such as fire, cyclones and grazing, are also key drivers of savanna structure and productivity which are in turn driven by rainfall patterns (Beringer et al., 2007; Bond et al., 2003; Hutley and Beringer, 2011; Hutley et al., 2013). Savanna grass productivity is very sensitive to rainfall and the biomass produced creates fodder for cattle and fuel for frequent burning, resulting in greenhouse gas emissions (Beringer et al., 1995, 2014; Moore et al., 2015, 2016). Therefore,
20  evaluating the relationship between inter-annual variation in rainfall and climate phenomena is crucial for predicting the responses of the water, energy and carbon cycles of savanna vegetation (Beringer et al., 2011a; Kanniah et al., 2013). Despite this there has been a paucity of research undertaken on the climatic influences of rainfall in northern Australia and savannas, however, there have been a number of continental scale analyses (e.g. Nicholls, 1989; Risbey et al., 2009; Schepen et al., 2012).

From north to south along the Northern Territory (NT), there is a substantial rainfall gradient (Table 1 and Fig. 1) known as the North Australian Tropical Transect (NATT). The north is highly seasonal with a characteristic tropical monsoonal climate (Bureau of Meteorology, 2011b; Hutley et al., 2011) and a rainy season between September and May (Nicholls et al., 1982; Suppiah and Hennessy, 1996), whereas the south is semi-arid to arid (Beringer et al., 2016) with very little seasonal variation
30  in rainfall (Hennessy et al., 1999). A sharp change in rainfall rate occurs around Dry Creek, with the stations north of Dry Creek experiencing higher rainfall, and the stations further south experiencing less rainfall.

surface-atmosphere exchanges (Hutley et al., 2011). Such transects can be used in novel ways to link climate with ecosystem function in two dimensions allowing for fairly simple statistical analysis to be performed. Hence, spatial change along a transect of a particular biome can be used to represent temporal change and has the potential to provide insight into how that landscape type may change in the future (Epstein et al., 2004). T, therefore these transect studies can therefore be an important tool for understanding the potential effects of temporal variability (both natural and anthropogenic).

Due to humanity's heavy demand for reliable and plentiful sources of water, combined with the threat of unknown future rainfall changes due to anthropogenic climate change, there is a need to improve our knowledge of precipitation drivers in order to improve future rainfall projections. The high importance of the Australian savanna landscapes to the global carbon and water balances illustrates the need to better understand the climatic processes in these regions. This paper determines how well different climate indices describe rainfall over the NT with a focus onalong the NATT. This has been achieved using spatial and temporal Pearson product-moment correlations including a brief examination of time lags. This research addresses correlation, not causation, over an extensive historical record (1900 to 2010). Trends in climate indices and rainfall over the NATT for this time period have also been examined. An unprecedented number of climate indices, representing climatic phenomena that are known to influence Australian rainfall variability, have been implemented in this research.

**2 Methodology**

**2.1 Site description**

Much of tThis paper is focussed onexamines the strong rainfall gradient along an extended version of the NATT, where 16 sites were chosen (Table 1 and Fig. 1) to examine the spatial relationships of rainfall with 16 different climate indices (Table 1 and Fig. 1). Due to the recognition of the importance of the NATT as a 'living laboratory' (Hutley et al., 2011) and the role of rainfall is ecosystem structure and function (Beringer et al., 2011b), a number of micrometeorological flux measurements have been established along the NATT as part of the regional flux network (OzFlux). Beringer et al. (2016) provide a description of the OzFlux network with initial cross site analysis. Howard Springs has been a long term monitoring site with observations initiated in 1996 (Eamus et al., 2001) and other sites include Fogg Dam (Beringer et al., 2013), Daly River (Hutley et al., 2011), Dry Creek (Beringer et al., 2011b) and Alice Springs (Cleverly et al., 2013). Rainfall decreases along the NATT from approximately 1,600 mm per year in the north, at a rate of ~200 mm per degree, to approximately 200 mm per year in the south, at a rate of ~100 mm per degree (Bureau of Meteorology, 2011a; Cook and Heerdegen, 2001) (Table 1 and Fig. 1). Associated with this decrease in precipitation is a change in vegetation structure and composition which varies from moist woodland savanna in the north to dry grasslands in the south (Beringer et al., 2011b; Hutley et al., 2011). Seasonality of rainfall also decreases from the north to the south, mostly due to the decrease in the influence of the monsoon moving further inland (Cook and Heerdegen, 2001).

**2.2 Climate indices**

WHere we used an extensive 110 years of spatial data, from 1900 to 2010, to help identify persistent long term correlations between precipitation and climate indices and to capture multiple events of each climatic phenomenon occurring at different frequencies. We also advanced on previous single index studies and used multipleby examining a vast number of climate indices (16) to assess the relative strength of influence of each climatic phenomenon, as described below and summarised in Table 2. The use ofUsing multiple climate indices enabled us to gain an insight into which understand the influence of different climatic phenomena may have the strongest relationships with onspatial rainfall patterns in the NT of rainfall, and ultimately which climatic phenomena may have antheir effect on ecosystem structure, function and distribution. A map showing the regions over which each climate index is calculated is shown in Fig. 2.

Indian Ocean and Indonesian phenomena

Climatic phenomena over the Indian Ocean and Indonesia affect are related to Australian precipitation both directly (e.g. Kamruzzaman et al., 2013) and indirectly by influencing other climatic phenomena (e.g. the Inter-decadal Pacific Oscillation and ENSO (Power et al., 1999)). SST anomaly (SSTA) data over these regions were used to calculate four climate indices as follows (Table 2 and Fig. 2). The IOD, represented by the *Dipole Mode Index* (DMI), is defined as the difference between the *Indian Ocean West Pole Index* (WPI, the average of the SSTAs over 50°E to 70°E and 10°N to 10°S) and the *Indian Ocean East Pole Index* (EPI, the average of the SSTAs over 90°E to 110°E and 0°N to 10°S (Saji et al., 1999)). Changes in the DMI coincide with changes in equatorial zonal wind variation (Saji et al., 1999). Anomalies in the DMI begin around June and increase until October when they reach a maximum, after which they quickly return to normal (Saji et al., 1999).

20  The *Indonesia Index* (II, the average of the SSTAs over 120°E to 130°E and 0°N to 10°S) characterises SSTAs over the Indonesian region and has been related to eastern Australian winter rainfall (Verdon and Franks, 2005) and NT rainfall (Schepen et al., 2012).

El-Niño Southern Oscillation

25  The relationship between ENSO and Australian rainfall is well known (e.g. Risbey et al., 2009; Ropelewski and Halpert, 1996). There are multiple ENSO indices of which we used six (Table 2 and Fig. 2). The *Southern Oscillation Index* (SOI) is commonly used in Australian rainfall studies (e.g. Risbey et al., 2009; Ropelewski and Halpert, 1996; Schepen et al., 2012; Suppiah and Hennessy, 1996) and is used to forecast and explain current rainfall patterns in Australia (Bureau of Meteorology, 2012b). The SOI for a given month is calculated as a function of MSLP difference between Tahiti and Darwin (Risbey et al., 2009) (Bureau of Meteorology, 2012).

30  The four *Niño indices* represent ENSO using SSTA measurements that are averaged over different regions of the Pacific Ocean (Fig. 2). These indices and their corresponding regions are; the Extreme Eastern Tropical Pacific SST (Niño 1+2), 90°W to 80°W and 0°N to 10°S; the Eastern Tropical Pacific SST (Niño 3), 150°W to 90°W and 5°N to 5°S; the East Central Tropical

Pacific SST (Niño 3.4), 170°W to 120°W and 5°N to 5°S; and the Central Tropical Pacific SST (Niño 4), 160°E to 150°W and 5°N to 5°S

(ESRL, 2012; Kamruzzaman et al., 2013; Risbey et al., 2009). Kamruzzaman et al. (2013) noted a seasonal pattern in Niño 1+2 and Niño 3 from 1957 to 2007.

5    -The *El Niño Modoki Index* (EMI) quantifies El Niño-Southern Oscillation Modoki events, which are similar to traditional ENSO events (Ashok et al., 2007), but with the . El Niño-Southern Oscillation Modoki events have maximum warming further east than normal ENSO events (
[revised manuscript text omitted]

**3.3.4 Lagged climate indices**

While this paper has so far focused solely on concurrent climate index-rainfall correlations, we briefly investigated the potential of lagged climate indices to improve rainfall description ability in the NT. Lagged SOI was found to produce stronger correlations than the initial SOI-rainfall correlations over some regions of the NT for certain time periods. Using monthly data, rainfall in the central-west and south-west of the NT showed the strongest correlations with SOI when rainfall was lagged behind SOI by three months (a positive lag), while the east of the NT favoured a concurrent correlation or positive one month lag (i.e. rainfall was lagged behind SOI, not shown). During summer, the highest rainfall correlations were shown with no lag in the north-east, a positive one month lag in the west, and a negative one to negative three month lag in the south (i.e. SOI was lagged behind rainfall by one to three months in the south). During autumn, stronger correlations with SOI were evident when lagged by between negative two to negative three months. Winter rainfall-SOI correlations were stronger in the southern half of the NT when lagged by positive three to positive four months whereas spring favoured a zero to positive two month lag in the south, a positive three month lag in the north east and a positive four month lag in the central west. It is therefore evident that while it may be possible to improve rainfall-climate index correlations using lagged data, due to the variation in the sign and number of months lagged, producing stronger correlations using lagged climate indices may not be a simple task.

**3.4 Implications**

[revised manuscript text omitted]